# Elevated synaptic vesicle release probability in synaptophysin/gyrin family quadruple knockouts

Mathan K Raja[1], Julia Preobraschenski[2], Sergio Del Olmo-Cabrera[3], Rebeca Martinez-Turrillas[1], Reinhard Jahn[2], Isabel Perez-Otano[1,3], John F Wesseling[1,3]*

[1]Department of Neuroscience, Universidad de Navarra, Pamplona, Spain; [2]Department of Neurobiology, Max Planck Institute for Biophysical Chemistry, Göttingen, Germany; [3]Institute for Neurosciences CSIC-UMH, San Juan de Alicante, Spain

**Abstract** Synaptophysins 1 and 2 and synaptogyrins 1 and 3 constitute a major family of synaptic vesicle membrane proteins. Unlike other widely expressed synaptic vesicle proteins such as vSNAREs and synaptotagmins, the primary function has not been resolved. Here, we report robust elevation in the probability of release of readily releasable vesicles with both high and low release probabilities at a variety of synapse types from knockout mice missing all four family members. Neither the number of readily releasable vesicles, nor the timing of recruitment to the readily releasable pool was affected. The results suggest that family members serve as negative regulators of neurotransmission, acting directly at the level of exocytosis to dampen connection strength selectively when presynaptic action potentials fire at low frequency. The widespread expression suggests that chemical synapses may play a frequency filtering role in biological computation that is more elemental than presently envisioned.

**Editorial note:** This article has been through an editorial process in which the authors decide how to respond to the issues raised during peer review. The Reviewing Editor's assessment is that all the issues have been addressed (see decision letter).

DOI: https://doi.org/10.7554/eLife.40744.001

*For correspondence: johnfwesseling@gmail.com

## Introduction

Synaptophysin 1 and 2 and synaptogyrin 1 and 3 constitute a major family of synaptic vesicle membrane proteins expressed widely, possibly in all synaptic vesicles throughout the animal kingdom (*Jahn et al., 1985*; *Stenius, 1995*; *Fernández-Chacón and Südhof, 1999*; *Takamori et al., 2006*). Synaptogyrin 2, also known as cellugyrin, is non-neuronal (*Janz and Südhof, 1998*). The widespread expression of neuronal family members suggests a fundamental role in synaptic transmission, but what that might be is not known.

Family members bind to the vSNARE synaptobrevin 2/VAMP 2, which is a core component of the machinery that catalyzes membrane fusion during synaptic vesicle exocytosis (*Söllner et al., 1993*; *Calakos and Scheller, 1994*; *Washbourne et al., 1995*; *Edelmann et al., 1995*; *Becher et al., 1999*; *Khvotchev and Südhof, 2004*). And, overexpression of family members potently inhibited neurotransmitter release in a cell line (*Sugita et al., 1999*). Despite this, the hypothesis that the native function might involve negative regulation of exocytosis has not been pursued extensively, possibly because no clear evidence was found for increases in neurotransmitter release at synapses from synaptophysin 1 and synaptogyrin 1 single and double knockouts (*McMahon et al., 1996*; *Janz et al., 1999*; *Abraham et al., 2006*; *Stevens et al., 2012*). Instead, recent research has been

focused on a variety of mechanisms that operate downstream of exocytosis, including: endocytosis of membrane; and/or recycling of proteins thought to be needed to catalyze subsequent rounds of exocytosis (*Kwon and Chapman, 2011*; *Gordon et al., 2011*; *Rajappa et al., 2016*). However, the previous studies, at least in mammals, involved exogenous expression or genetic deletion of synaptophysin 1 and synaptogyrin 1, whereas possible compensatory activity of synaptophysin 2 and synaptogyrin 3 has never been assessed (*McInnes et al., 2018*).

Here, we report that individual action potentials trigger exocytosis of a higher fraction of the readily releasable vesicles at a variety of synapse types from quadruple knockout mice (QKO) where all four neuronal family members have been deleted (see Materials and methods). No deficit was detected in other presynaptic parameters that control function such as the capacity of the readily releasable pool (RRP) for storing vesicles and the timing of vesicle recruitment to the RRP during light or heavy use. The results suggest strongly that family members play an inhibitory role at the level of exocytosis rather than the facilitatory downstream role that is currently envisioned. A follow-on analysis of double and triple knockouts showed that synaptophysin 1 and synaptogyrin 3 can compensate for missing family members, whereas synaptophysin 2 seemed to play a dominant negative role.

## Results

QKO mice appeared to develop normally when housed in individually ventilated cages, and produced litters of normal size. However, adults were prone to convulsions, sometimes causing death, especially after being startled. We were not able to maintain the colony in a second facility where ventilated cages were not available, suggesting that the ventilation system aided survival, possibly by producing continuous white noise that limited startling. A quantitative western blot analysis of homogenized tissue and purified synaptosomes from QKO brains revealed a selective decrease in VAMP 2 levels, but no major changes in a wide variety of other synaptic proteins (*Figure 1*); a decrease in VAMP 2 levels was detected previously in synaptophysin 1 single knockouts (*McMahon et al., 1996*).

### Elevated neurotransmitter release at calyx of Held synapses from QKO mice

In a first set of experiments to determine the primary functional deficit, we found that calyx of Held synapses from QKO mice were substantially stronger than WT when action potentials were evoked in the afferent axon at low frequency (*Figure 2A–B*). Excitatory postsynaptic currents (EPSCs) recorded in voltage clamped principal neurons of the medial nucleus of the trapezoid body (MNTB) had a greater quantal content (*Figure 2C*), indicating that the synapses were stronger because of exocytosis of transmitter from more presynaptic vesicles, in-line with the presynaptic locus of expression of synaptophysin family members. Spontaneous quantal release was elevated by a similar amount (*Figure 2D–E*), and the size of quantal events was elevated by a smaller amount (*Figure 2F*). No significant alterations were detected in the time courses of the EPSCs evoked with low-frequency stimulation or in spontaneous events (*Figure 2—figure supplement 1*).

### Elevated probability of release, with no alteration in RRP size

The number of vesicles that undergo exocytosis when action potentials are fired at low frequency is determined by two factors that seem to be controlled independently: the number of vesicles within a readily releasable pool, termed *RRP content* and possibly determined by the number of sites in the active zone area of the plasma membrane where vesicles dock; and the mean probability of release per vesicle ($\bar{p}_v$) within the RRP (*Figure 3A*, left); the abbreviation $P_r$ has been used to denote the same concept in some other studies but is also sometimes used instead to denote the probability of release per *synapse*, which is the mathematical product of $\bar{p}_v$ and RRP content, and thus a different concept. To determine which of the parameters was altered, we stimulated at 300Hz for 300ms (90 action potentials; *Figure 3B*). The difference in synaptic strength disappeared quickly, by the 4th action potential (*Figure 3B–C*), and there was no difference in the total number of quanta released during the first 150ms of the trains (*Figure 3D*). The result suggests that the RRP content was not altered at QKO synapses because 150ms is enough to nearly completely exhaust the RRP (*Mahfooz et al., 2016*).

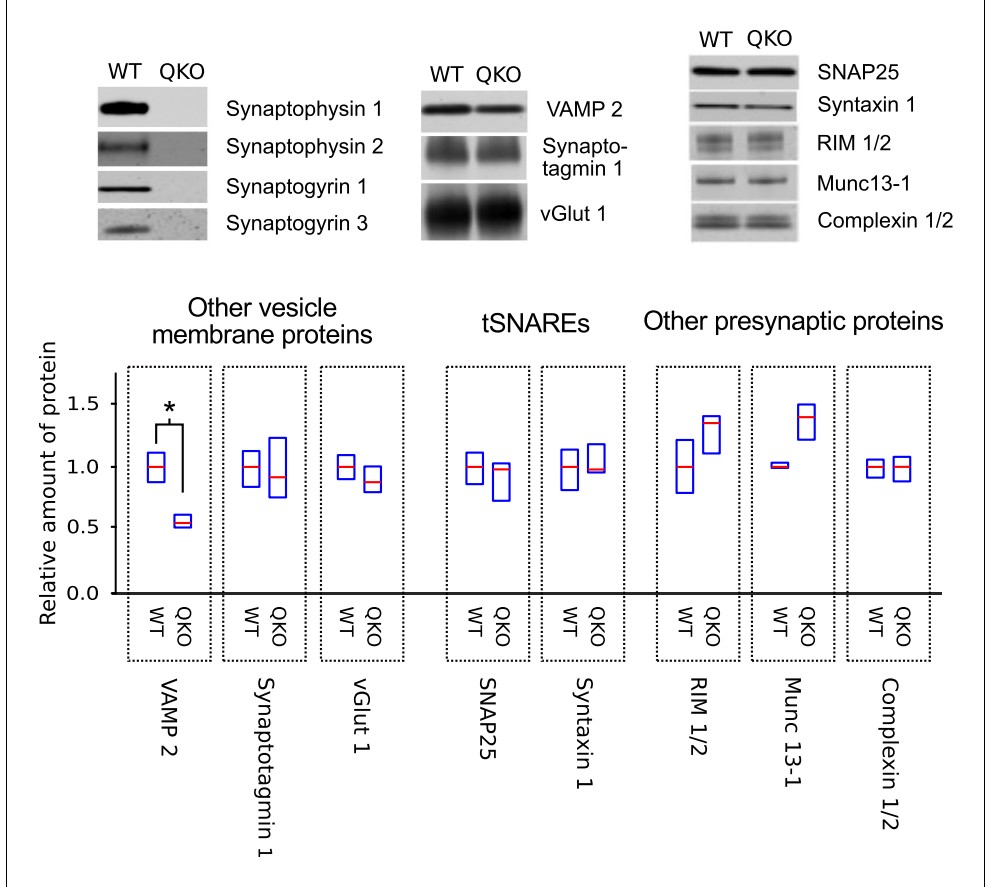

**Figure 1.** Selective decrease in VAMP 2 levels in synaptosomes of QKO mice. Representative immunoblots and quantification of the indicated proteins from synaptosomes purified from brains of 3-month-old WT and QKO mice. Synaptosomes were prepared separately from cohorts of four male and four female individuals, but results were pooled because no substantial differences were detected between sexes. Horizontal lines are median values, boxes are the middle two quartiles. *p<0.05; Wilcoxon rank sum with Bonferonni correction for multiple comparisons; n ≥ 6 (2 independent preparations; samples were run at least 3 times).
DOI: https://doi.org/10.7554/eLife.40744.002
The following figure supplement is available for figure 1:

**Figure supplement 1.** More extensive biochemical analysis including more proteins, different ages, and different methods for sample preparation.
DOI: https://doi.org/10.7554/eLife.40744.003

Technically, the number of quanta released during trains that exhaust the RRP is not a perfect measure of RRP content because new vesicles are continually recruited and contents released during ongoing stimulation (*Figure 3A*, right). However, the amount of recruitment can be estimated by a variety of methods; two that have been proposed for this purpose are plotted in *Figure 3D* (i.e. curves marked 'S' and 'M'). To our knowledge, the two span the full range of quantitative models that have been proposed; theory 'S' was proposed earlier, by *Schneggenburger et al. (1999)*, whereas 'M' incorporates the conclusion of *Mahfooz et al. (2016)* that the RRP has a fixed capacity and vesicles are recruited to vacant spaces, such as empty release sites, as illustrated in *Figure 3A* (right panel). Although the various methods produced a variety of estimates for the amount of recruitment during the trains, all methods agreed that the amount was not different at QKO compared to at WT synapses (e.g. intercepts of curves 'S' and 'M' with ordinate-axis in *Figure 3D*). This result confirms that the initial RRP content was not altered at QKO synapses.

In contrast, $\bar{p}_v$ is calculated by dividing the number of quanta released after isolated action potentials by the RRP content, and was approximately double at QKO synapses (*Figure 3E*). These results

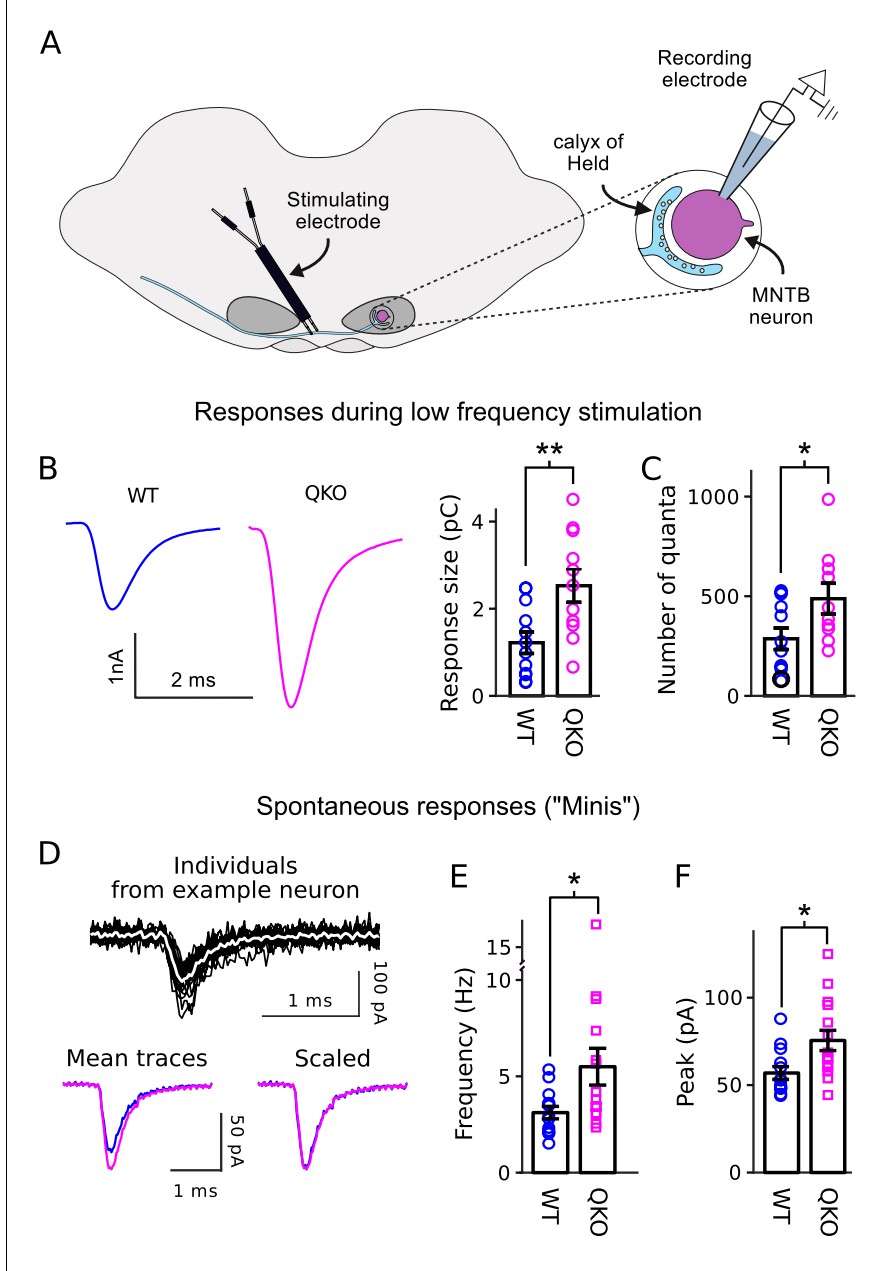

**Figure 2.** Increased transmitter release at QKO calyces of Held. (**A**) Diagram of calyx of Held preparation; MNTB is the medial nucleus of the trapezoid body. (**B**) Larger responses at QKO synapses after isolated/low frequency presynaptic action potentials (*i.e.*, each after at least 1 min of rest). Traces are average responses across all preparations; n ≥ 11 calyces for both WT and QKO, each from a separate slice; experimenter was blind to genotype; extracellular 1mM kynurenic acid was used throughout. (**C**) Response sizes from (**B**) after normalizing by quantal size calculated as in *Mahfooz et al. (2016)*. (**D-F**) Analysis of spontaneous responses recorded before adding kynurenic acid; n ≥ 13 calyces for both WT and QKO. (**D**) Black traces are overlay of all individuals from a single QKO neuron. The white trace is the mean that was used later for quantification. Mean traces are means of all individuals across all preparations. (**E** and **F**) Data points correspond to single preparations. Bars are mean ± s.e.m.; *p < 0.05; **p < 0.01; Wilcoxon rank sum.

DOI: https://doi.org/10.7554/eLife.40744.004

The following figure supplement is available for figure 2:

**Figure supplement 1.** No difference in shape of EPSC at QKO calyces of Held.

DOI: https://doi.org/10.7554/eLife.40744.005

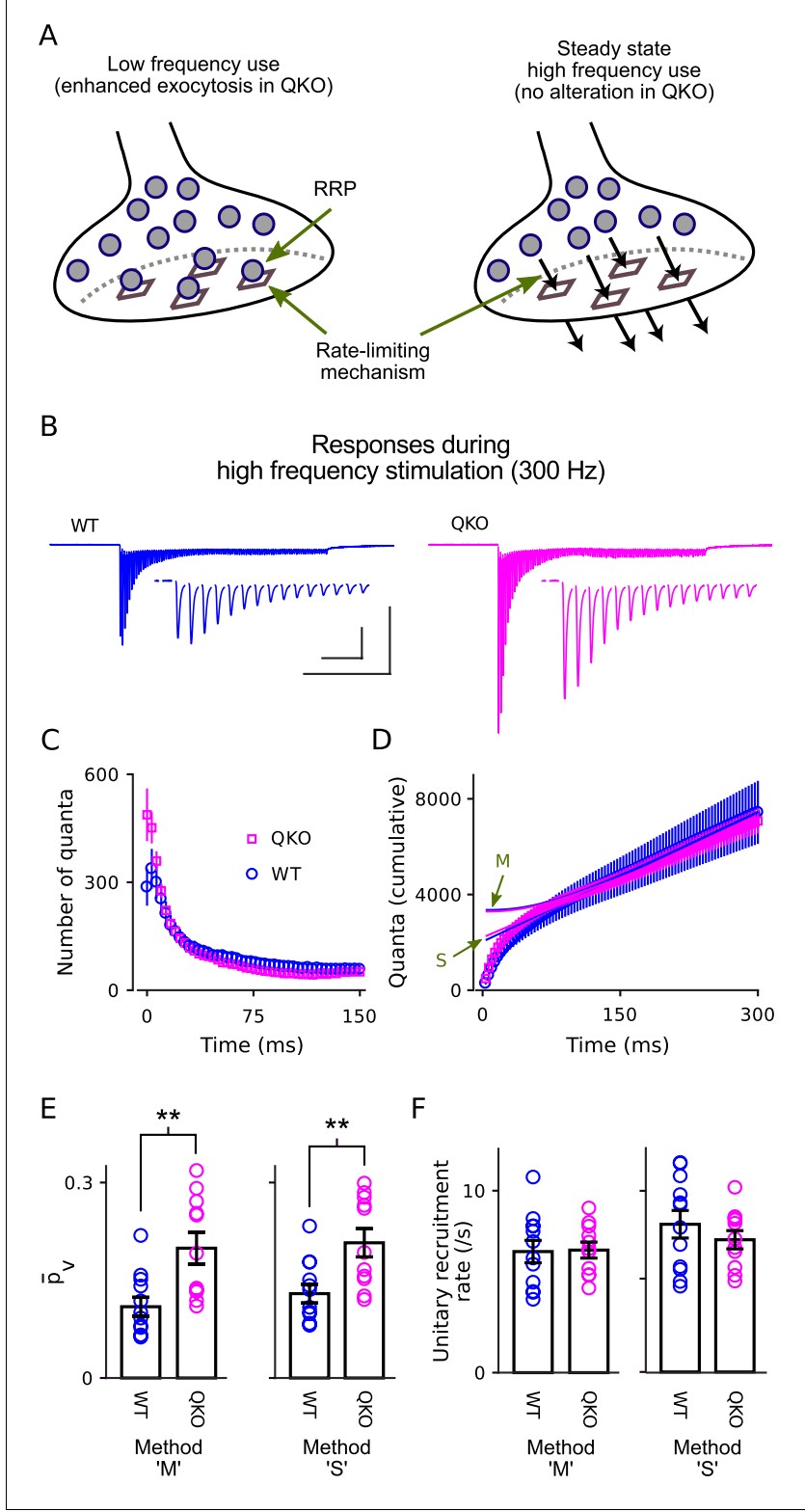

**Figure 3.** Selective increase in probability of release at QKO calyces of Held. (**A**) Diagram illustrating the RRP when nearly full during low-frequency stimulation (left) and when driven to a near-empty steady state by high-frequency stimulation (right). Gray circles represent vesicles, vesicles docked to release sites (squares) are readily releasable. By definition, the quantal content of individual synaptic responses is equal to the mean probability of release per vesicle within the RRP - $\bar{p}_v$ - multiplied by the number of vesicles within the RRP. However, the precise

*Figure 3 continued on next page*

*Figure 3 continued*

value of $\bar{p}_v$ ceases to be relevant when the RRP is driven to a near-empty steady state because vesicles undergo exocytosis soon after being recruited and, as a consequence, recruitment to the RRP (black arrows, right panel) supplants vesicle exocytosis as the rate-limiting mechanism. (**B**) Average response across all calyces during 300 ms of 300 Hz stimulation after blanking stimulus artifacts. Scale bars are 1 nA *vs* 100 ms (outer) and 1 nA *vs* 10 ms (inner, corresponding to the insets showing first 15 responses). (**C**) Mean number of quanta for each response *vs* time. Responses were first measured as the current integral after subtracting a baseline calculated from the 100 ms before stimulation began, and then normalized by mean quantal size. (**D**) Cumulative number of quanta. Theoretical curves are estimates of the cumulative response generated by release of transmitter recruited to the RRP during ongoing stimulation plus the offset needed to make the phase between 150 ms and 300 ms match the cumulative release; the value at *Time = 0* equals the capacity of the RRP for storing vesicles. Lines marked 'S' are calculated using the method in *Schneggenburger et al. (1999)*, whereas lines marked 'M' are calculated using Eqn (1) in *Mahfooz et al. (2016)*, and describe the model illustrated in (**A**); note that both curves for both genotypes are plotted (magenta for QKO, blue for WT). The full 300 ms of 300 Hz stimulation elicited multiple rounds of exocytosis of readily releasable vesicles for both genotypes, including a total of 7076 ± 531 quanta at QKO synapses, which is more than double even the largest estimates of RRP content (ordinate intercept of 'M'). (**E**) $\bar{p}_v$ for calyces calculated using the theories in (**D**) to estimate vesicle recruitment (p < 0.01; rank sum; same preparations as *Figure 2*). (**F**) Unitary recruitment rate for individual calyces. The unitary recruitment rate is defined as the fraction of vacant space within the RRP replenished in a given amount of time; the concept is depicted by the black arrows in (**A**), right, and is analogous to a rate constant in first-order kinetics.

DOI: https://doi.org/10.7554/eLife.40744.006

The following figure supplement is available for figure 3:

**Figure supplement 1.** Controls for receptor desensitization.

DOI: https://doi.org/10.7554/eLife.40744.007

show that removing synaptophysin family members increased the value of $\bar{p}_v$ without altering RRP content. Taken together, they suggest that endogenous synaptophysin family members inhibit neurotransmission downstream of vesicle recruitment to the RRP.

## No alteration in the timing of vesicle priming

The conclusion is consistent with early experiments where exogenous expression of family members inhibited exocytosis (*Sugita et al., 1999*). However, one of the current hypotheses is that at least synaptophysin 1 plays a post-exocytosis role in clearing components of spent vesicles from the release machinery. The idea is that clearance determines the rate at which new vesicles can be recruited to the RRP during heavy use (*Kwon and Chapman, 2011*; *Gordon et al., 2011*; *Rajappa et al., 2016*). If so, removing family members would have produced a deficit in the rate at which vesicles are recruited to the RRP at later times during the trains of 300 Hz stimulation, which drove multiple rounds of exocytosis (see Legend of *Figure 3D*), and during subsequent rest intervals. But no such deficits were seen at QKO synapses.

That is, the methods used above to estimate the total amount of vesicle recruitment during trains additionally produce estimates of the ongoing rate of recruitment during heavy use (arrows in *Figure 3A*, right). All produced matching estimates for QKO and WT synapses (*Figure 3F*); see *Figure 3—figure supplement 1* for control experiments verifying that the analyses were not confounded by postsynaptic mechanisms such as glutamate receptor desensitization.

The absence of a deficit in recruitment during extended stimulation could additionally be deduced without referencing any of the methods simply from the observation that steady state quantal output after the 150th ms of 300 Hz stimulation was similar at QKO synapses compared to WT (53.4 ± 4.7 quanta/action potential vs 59.2 ± 11.7). Nor did we find any alteration in the timing of RRP replenishment during rest intervals that followed 300 Hz stimulation (*Figure 4*). These results do not support the hypothesis that synaptophysin family members play a critical post-exocytosis role in facilitating ongoing vesicle priming and release, at least at calyx of Held synapses.

## Elevated $\bar{p}_v$ pertains to both high and low $p_v$ vesicles

The results so far suggest that synaptophysin family members ordinarily play an inhibitory role in neurotransmission downstream of vesicle recruitment to the RRP and upstream of neurotransmitter

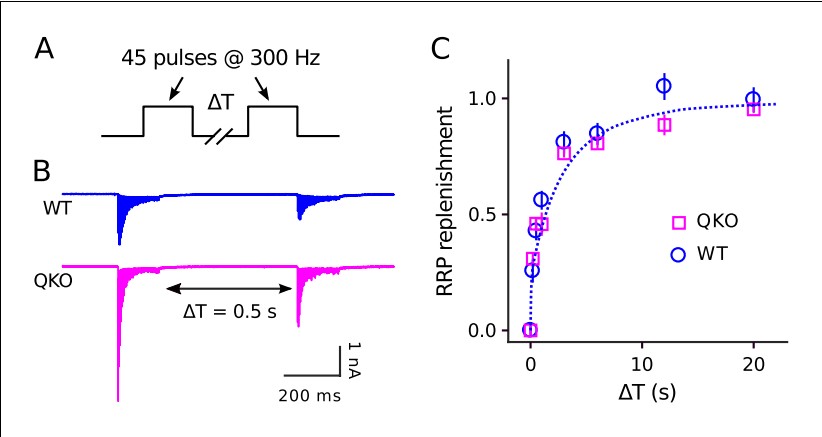

**Figure 4.** No alteration in timing of RRP replenishment at QKO calyces of Held. (**A**) Experimental design. (**B**) Averaged recordings for trials with a rest interval of 0.5s between trains from single preparations. (**C**) RRP replenishment vs time estimated as in *Mahfooz et al. (2016)*; n ≥ 20 trials from 7 calyces for QKO and 8 trials from 3 calyces for WT. The dashed line is $RRP_t = 1 - e^{-\int \hat{\alpha}_t}$ with $\hat{\alpha}_t$ the decaying exponential defined by Eqn (4) in *Mahfooz et al. (2016)*, except $\hat{\alpha}_0 = 5.4/s$ to match the value used to generate curve 'M' in *Figure 3D*.
DOI: https://doi.org/10.7554/eLife.40744.008

release, consistent with the possibility that the action is directly at the level of exocytosis. However, some current models include intervening priming steps between recruitment and exocytosis.

That is, the initial idea was that the RRP is a homogeneous pool (*Elmqvist and Quastel, 1965*; *Vere-Jones, 1966*), but there is now widespread agreement that some readily releasable vesicles are released more slowly than others during repetitive stimulation owing to a lower probability of release (*Wu and Borst, 1999*; *Sakaba and Neher, 2001*; *Moulder and Mennerick, 2005*). For clarity, we refer to the readily releasable vesicles with high and low probability of release as high and low $p_v$ vesicles, respectively, but note that it is possible that there are more than only two classes (*Mahfooz et al., 2016*; *Taschenberger et al., 2016*); elsewhere, low $p_v$ vesicles have been termed slow-releasing, or reluctantly releasable.

Both high and low $p_v$ vesicles seem to be immediately releasable, at least in the calyx of Held preparation examined here, and the timing of exocytosis is tightly synchronized to action potentials (*Mahfooz et al., 2016*). Our own models continue to treat recruitment to the RRP as the final vesicle trafficking step upstream of exocytosis (*Mahfooz et al., 2016*). the idea is that low and high $p_v$ vesicles are docked to distinct types of release sites (*Figure 5A*; *Hu et al., 2013*; *Müller et al., 2015*; *Böhme et al., 2016*). However, recent models of other research groups include additional mechanisms where individual vesicles already within the RRP reversibly transition between a variety of primed states distinguished by a range of release probabilities (*Lee et al., 2013*; *Neher, 2017*). If so, synaptophysin family members might either: limit the rate of transition of one of the forward steps, decreasing the fraction of vesicles in a high $p_v$ state; or directly inhibit exocytosis of low and high $p_v$ vesicles alike.

To determine whether the action is at the level of exocytosis or upstream, we conducted frequency jump experiments where high $p_v$ vesicles are first eliminated from the RRP with submaximal stimulation of 50 or 100 Hz, leaving a standing population of low $p_v$ vesicles that can then be released by subsequent 300 Hz stimulation (*Figure 5A*). The experimental design was developed previously for isolating the kinetics of release of low $p_v$ vesicles at WT calyces of Held (*Mahfooz et al., 2016*). In the present case, the full experiment consisted of 6 types of interleaved trials: frequency jumps where 1000 ms of 50 Hz stimulation was followed by 200 ms of 300 Hz stimulation and a matched control where the stimulating frequency was maintained at 50 Hz throughout (*Figure 5B–C*); two additional types of frequency jumps where 500 ms or 750 ms of 100 Hz stimulation was followed by 200 ms of 300 Hz stimulation, along with a matched control where the stimulating frequency was maintained at 100 Hz throughout (*Figure 5—figure supplement 1A*); and an

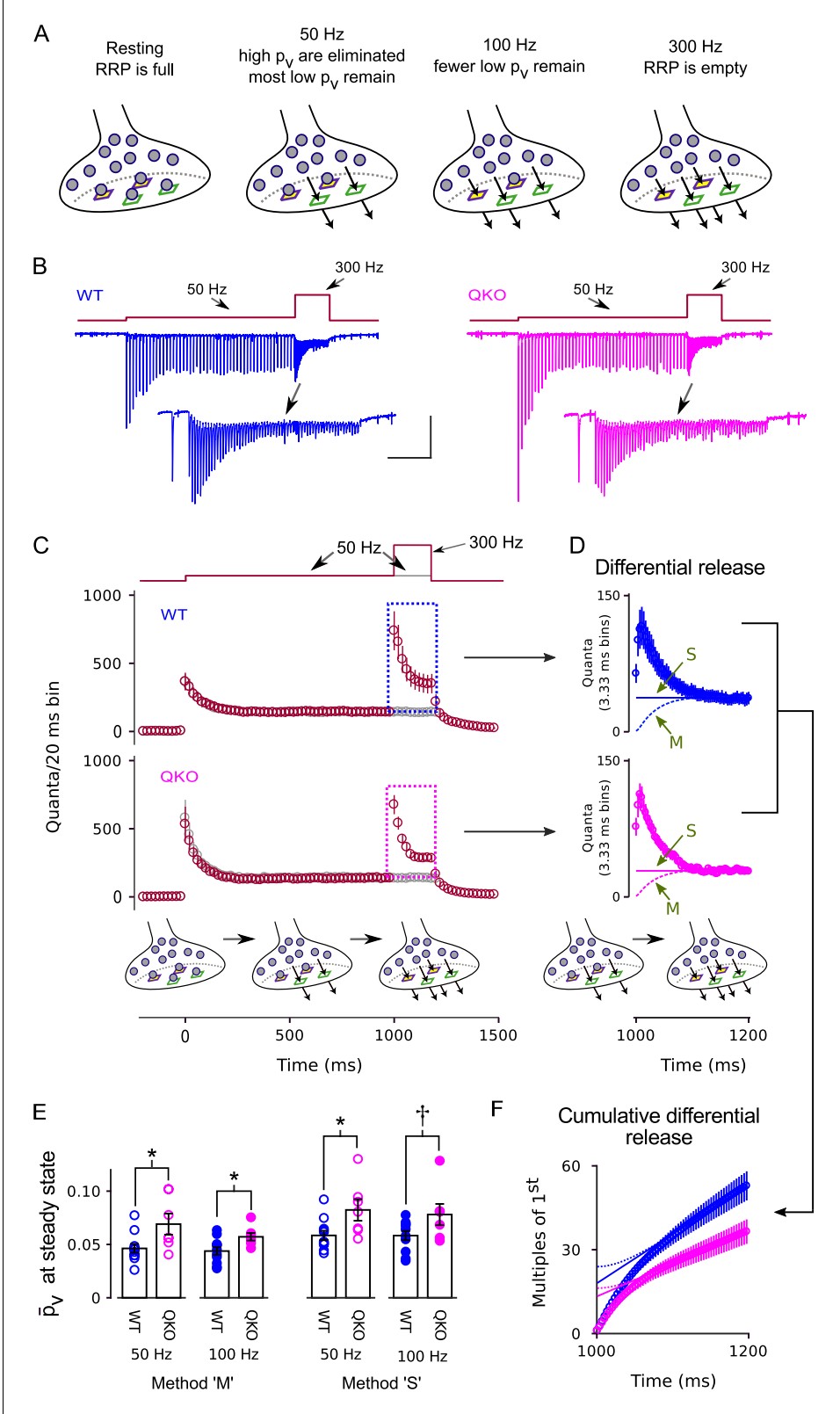

**Figure 5.** Enhanced release of vesicles with low $p_v$ remaining in RRP after 50 Hz or 100 Hz stimulation at QKO synapses. (**A**) Diagram illustrating the steady state contents of the RRP for a range of stimulating frequencies. Release sites are depicted as stable, and are characterized by either a high (green squares) or a low (purple squares with yellow interior) probability of catalyzing the exocytosis of docked vesicles (i.e. high or low $p_v$ release

*Figure 5 continued on next page*

*Figure 5 continued*

sites). Alternative models where the readily releasable vesicles transition back and forth between high and low $p_v$ states would be the same except the release sites would not have a defined $p_v$ when empty, and the locations of the high and low $p_v$ vesicles would change over time. In either case, 50 Hz and 100 Hz stimulation is rapid enough to eliminate the vesicles with high $p_v$ from the RRP, but leaves a flow through pool of low $p_v$ vesicles that can then be released by subsequent 300 Hz stimulation. (B) Example recordings for trials where the stimulation frequency was increased from 50 Hz to 300 Hz; blue is WT, magenta is QKO. The insets are the last response during 50 Hz stimulation and responses during subsequent 300 Hz stimulation. The scale bars pertain to both sets of traces: the vertical bar is 1 nA and the horizontal is 250 ms for the full traces; and 500 pA and 50 ms for the insets. (C) Mean responses for the full data set quantified as the number of quanta released during sequential 20 ms segments, allowing direct comparison of the time-averaged rate of release when stimulation was 50 vs when 300 Hz; single segments contain the quantal content of responses to single action potentials for times when the stimulating frequency was 50 Hz, and of responses to 6 consecutive action potentials when the frequency was 300 Hz. Red symbols are for trials where stimulation was increased from 50 to 300 Hz, and gray are for trials where stimulation was maintained at 50 Hz throughout, as diagrammed at top. Boxes demarcate responses used to calculate the differential release in (D). (D) The additional release - termed differential release here - elicited by increasing the stimulation frequency to 300 Hz was calculated by subtracting the time-averaged values during continued 50 Hz stimulation from the corresponding values during 300 Hz stimulation. The theoretical curves are estimates of the fraction of the differential release produced by exocytosis of neurotransmitter that was recruited to the RRP during ongoing stimulation. 'M' and 'S' signify the same as in *Figure 3D*; 'M' describes the model in the illustrations at bottom and in (A). (E) $\bar{p}_v$ values for individual calyces during steady state 50 and 100 Hz stimulation using the same methods 'M' and 'S' used to generate the theoretical curves in (D) († is p < 0.1, * is p < 0.05; n ≥ 7; rank sum). (F) Cumulative version of the plot in (D), except after normalizing by the quantal content of the first differential response. Solid and dashed lines represent theories 'M' and 'S' as in (D) except offset to match the rightmost data points. Without the normalization, the lines would intersect the y-axis at the value that corresponds to the contents of the RRP at the start of 300 Hz stimulation (*Schneggenburger et al., 1999*). However, the normalization converts the estimate into multiples of the quantal content of the first response, making the intersection equal to $1/\bar{p}_v$.
DOI: https://doi.org/10.7554/eLife.40744.009

The following figure supplements are available for figure 5:

**Figure supplement 1.** Frequency jumps from $100\,\mathrm{Hz}$.
DOI: https://doi.org/10.7554/eLife.40744.010
**Figure supplement 2.** Cumulative plots for full experiment.
DOI: https://doi.org/10.7554/eLife.40744.011

additional control where stimulation was 300 Hz for 300 ms, with no prior stimulation for at least 1 min (subset of data in *Figures 2–3*).

*Figure 5B–C* and *Figure 5—figure supplement 1A* show that 300 Hz stimulation transiently increased the rate of release following 50 Hz or 100 Hz stimulation, extending to QKO calyces of Held the previous finding that 100 Hz stimulation is not sufficiently frequent to exhaust the RRP at WT synapses (*Mahfooz et al., 2016*). To determine the $\bar{p}_v$ value for the vesicles remaining in the RRP, we first isolated the increase in release by subtracting the time-averaged steady state response recorded during the matched trials when the stimulation frequency was not increased (gray data points in *Figure 5C* and *Figure 5—figure supplement 1A*; the amount of increase - termed the differential release - is plotted in *Figure 5D* and *Figure 5—figure supplement 1B*). The value for $\bar{p}_v$ was then calculated by dividing the differential release after the first action potential during 300 Hz stimulation by the RRP content at the start of 300 Hz stimulation. The RRP content at the start of 300 Hz stimulation was estimated by subtracting the amount of recruitment to the RRP during 300 Hz stimulation from the total.

As above for *Figure 3D–E*, the estimate for the amount of recruitment during 300 Hz stimulation depended partly on assumptions about mechanism that continue to be debated (lines marked 'M' *vs* 'S' in *Figure 5D* and *Figure 5—figure supplement 1B*). However, all methods produced $\bar{p}_v$ values for the vesicles remaining in the RRP that were higher for QKO synapses compared to WT (*Figure 5E*). The effect can be seen most clearly in the cumulative plots of the differential release measurements in *Figure 5F*, which are normalized so that the theoretical curves produced by the 'M' and 'S' theories intersect the y-axis at $\frac{1}{\bar{p}_v}$ (see Legend). Note that the effect is not as readily apparent in *Figure 5C–D* because the standing state fullness of the RRP was lower at QKO synapses

(*Figure 5—figure supplement 2*) - which is an expected consequence of the higher $\bar{p}_v$ value (*Mahfooz et al., 2016*) - and because of the large bin size in *Figure 5C*.

The $\bar{p}_v$ values for both WT and QKO were approximately 3-fold lower during 50 or 100 Hz stimulation compared to the corresponding values when the RRP was full (compare *Figure 5E* to *Figure 3E*), confirming that high $p_v$ vesicles were eliminated. And indeed, almost all of the vesicles remaining within the RRP during both 50 and 100 Hz stimulation must have been in the low $p_v$ state because 100 Hz stimulation depleted the RRP to a greater extent (*Figure 5—figure supplement 2*), but did not further lower the value for $\bar{p}_v$ (*Figure 5E*).

These results do not distinguish between the classes of models with single and multiple sequential vesicle priming steps mentioned above, but do indicate that synaptophysin family members inhibit exocytosis of low and high $p_v$ vesicles alike. The results therefore strongly suggest that family members inhibit neurotransmission downstream of the final step in vesicle priming, which is consistent with the increase in spontaneous exocytosis seen in the absence of action potentials documented in *Figure 2D–E*. The analysis is based on the premise that low $p_v$ vesicles are immediately releasable, which seems likely because release of neurotransmitter continues to be tightly synchronized to action potentials during frequency jumps that are initiated after the high $p_v$ vesicles have been eliminated, and because synapses with more low $p_v$ vesicles express more paired pulse facilitation when the RRP is full (*Figure 5B* and *Figure 5—figure supplement 1C*; *Mahfooz et al., 2016*). Nevertheless, we did additionally considered the alternate scenario where the low $p_v$ vesicles are not immediately releasable (*Miki et al., 2016*; *Gustafsson et al., 2019*), but found it was not compatible with the results in *Figure 5D* and *Figure 5—figure supplement 1B*, as explained in the Legend of *Figure 5—figure supplement 1*.

## Elevated $\bar{p}_v$ at Schaffer collateral synapses from QKO mice

We next conducted experiments analogous to *Figures 3–4* on synapses between Schaffer collaterals and CA1 pyramidal neurons of the hippocampus. We stimulated with 20 Hz trains instead of 300 Hz because 20 Hz is frequent enough to nearly completely empty the RRP owing to ~20-fold slower recruitment of new vesicles during ongoing stimulation compared to the calyx of Held (*Wesseling et al., 2002*; *Mahfooz et al., 2016*).

Estimating the quantal content of responses from individual Schaffer collaterals is not as straightforward as for the calyx of Held because the number of afferent axons activated during a typical experiment is an unknown function of the strength of the individual pulses of stimulation (*Figure 6A*). Nevertheless, similar to at the calyx of Held, postsynaptic responses depressed more rapidly at QKO synapses compared to WT (*Figure 6B–C*). A kinetic analysis indicated that $\bar{p}_v$ was approximately double (*Figure 6D–E*), whereas the timing of vesicle recruitment to the RRP during ongoing stimulation was not altered (*Figure 6F*). And, no difference was detected in the time course of RRP replenishment during subsequent rest intervals (*Figure 6G*). These results extend to hippocampal synapses the conclusion that the machinery that catalyzes synaptic vesicle exocytosis becomes more efficient after removing synaptophysin family proteins, whereas the timing of vesicle recruitment to the RRP is not altered. The comparison between calyces of Held and Schaffer collateral synapses is a good test for generality across synapse types because, in nature, Schaffer collateral synapses are typically used at frequencies that are ~15-fold lower in addition to striking morphological and molecular differences and the ~20-fold difference in the timing of vesicle trafficking already noted above (*Ranck, 1973*; *Hermann et al., 2007*; *Borst and Soria van Hoeve, 2012*).

## Higher throughput assay in primary cell culture

To confirm that synaptic vesicle exocytosis is increased at QKO hippocampal synapses when action potentials are fired at low frequency, and to assess the contribution of each of the four synaptophysin family members, we then developed an optical imaging assay in primary cell culture with higher throughput than the electrophysiological assays (see *Figure 7A* and diagram atop *Figure 7B*). We first loaded the recycling synaptic vesicles with FM4-64 dye during 60 s of 20 Hz electrical stimulation (*Gaffield and Betz, 2006*; see *Figure 7—figure supplement 1* for example images). We then monitored destaining with time-lapse fluorescence imaging during low frequency (0.2 Hz) stimulation in the absence of dye, followed by near complete destaining with a second 20 Hz train.

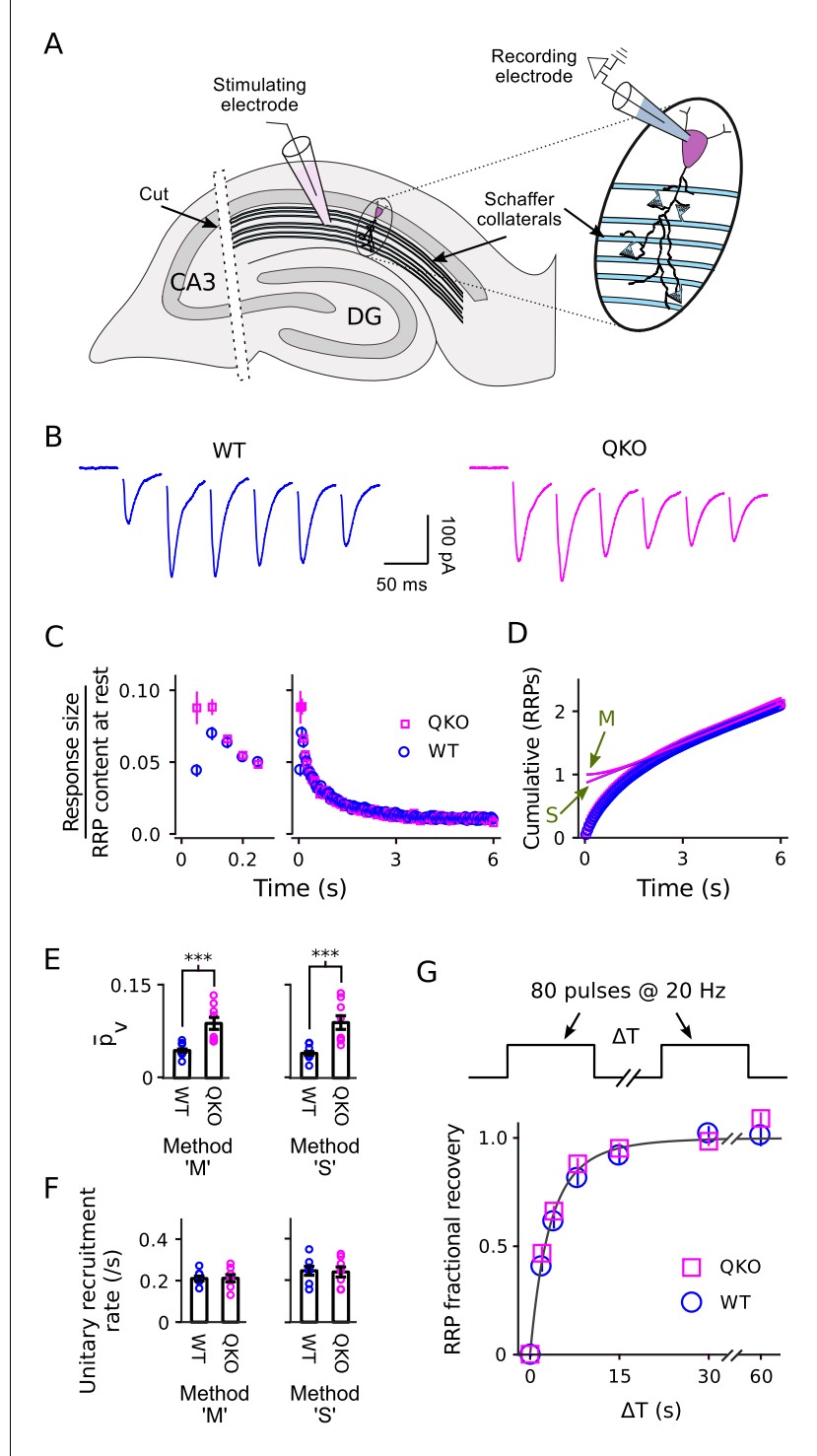

**Figure 6.** Electrophysiological analysis of Schaffer collateral synapses. (**A**) Diagram of ex vivo hippocampal slice preparation. (**B-F**) Increased $\bar{p}_v$ at QKO synapses, but no differences in timing of vesicle trafficking during ongoing stimulation. (**B**) Responses during 20 Hz stimulation; shown are the first 300 ms of 6 s-long trains from individual preparations; traces are the average of four trials. (**C**) Mean sizes of responses *vs* time. Individual trials were repeated at least three times for each preparation, and each preparation was allowed to rest at least 4 min before beginning each trial (n = 8 preparations per genotype). Responses were measured as the current integral and then normalized by the RRP contents at the start of stimulation calculated as in *Wesseling et al. (2002)*; when normalized this way, the leftmost values are then equal to $\bar{p}_v$. (**D**) Cumulative responses, normalized as in (**C**).

*Figure 6 continued on next page*

*Figure 6 continued*

Theoretical curves are labeled as in *Figure 3D*, except here 'M' refers to the theory described in *Wesseling et al. (2002)*, which is analogous to *Mahfooz et al. (2016)* but specific for hippocampal synapses. (E) $\bar{p}_v$ values across preparations (p < 0.001; rank sum). Methods 'M' and 'S' are the same as for *Figure 3E*. (F) Values for the unitary recruitment rate across preparations. (G) RRP replenishment *vs* time; the dashed line is $RRP_t = 1 - e^{-\int \alpha_t}$ with $\hat{\alpha}_t$ the decaying exponential $\alpha(t)$ in *Wesseling et al. (2002)*.

DOI: https://doi.org/10.7554/eLife.40744.012

We reasoned that destaining would be directly proportional to $\bar{p}_v$ during low-frequency stimulation because the RRP would remain almost completely full. In contrast, we reasoned that destaining would not be influenced at all by $\bar{p}_v$ during 20 Hz stimulation which empties the RRP in less than the 4 s interval between acquisition of successive images, after which transmitter release is no-longer influenced by $\bar{p}_v$ and is instead rate-limited by vesicle recruitment to the RRP (*Wesseling et al., 2002*; see *Figure 7A*).

And indeed, QKO synapses destained almost 2-fold more than WT during the 0.2 Hz stimulation (*Figure 7A–C*), confirming that $\bar{p}_v$ is elevated. And, no differences were detected during subsequent 20 Hz stimulation (not shown), or during 20 Hz stimulation when the 0.2 Hz train was omitted (*Figure 7D*), confirming that the timing of vesicle recruitment to the RRP was not altered. Furthermore, no differences between QKO and WT were detected in the amount of staining during loading, further supporting the conclusion of no alterations in rate-limiting steps in vesicle trafficking in QKO synapses.

## Analysis of triple and double knockouts

Of the triple knockouts lacking all but one of the neuronal family members, synaptophysin 1 or synaptogyrin 3 alone largely compensated for the loss of the other three family members (*Figure 7C*).

Intriguingly, synapses from triple knockouts expressing only synaptophysin 2 were not noticeably different from QKO, but the combination of synaptophysin 2 with either synaptophysin 1 or synaptogyrin 3 was less effective at compensation than synaptophysin 1 or synaptogyrin 3 alone (*Figure 7E*). This result suggests that synaptophysin 2 may act as a competitive inhibitor of the function of other family members, or play a dominant negative role. And indeed, synaptophysin 2 is unique in that it lacks many of the sites for C-terminal tyrosine phosphorylation that are striking features of the other family members (*Evans and Cousin, 2005*).

## Discussion

Synaptophysin family proteins are widely expressed in synaptic vesicle membranes, with more individuals per vesicle than the extensively studied synaptotagmins, although likely less than the vSNARE VAMP 2 and homologs (*Takamori et al., 2006*; *Wilhelm et al., 2014*). Despite the abundance, information about the role in presynaptic function has been elusive. Here, we show that the efficiency of the release machinery is elevated at a variety of synapse types from knockout mice where all four neuronal family members have been deleted, with no indication of any alteration in RRP content or the timing of vesicle recruitment to the RRP during light or heavy use. The new results strongly suggest that synaptophysin family members modulate function at the level of exocytosis. If so, the native action likely includes inhibition, especially when the results are taken together with an earlier study where exogenous synaptophysin 1 and synaptogyrin 1 potently inhibited exocytosis (*Sugita et al., 1999*; but see *Alder et al., 1995*).

The concern that the elevated $\bar{p}_v$ seen at QKO synapses might instead reflect complicating developmental or compensatory mechanisms that are not directly related to the native function of the missing proteins is countered by the following. First, the alteration is unusually specific; $\bar{p}_v$ was elevated without desynchronizing the relationship between action potentials and neurotransmitter release, or disrupting other vesicle trafficking mechanisms that control presynaptic function such as the capacity of the RRP for storing synaptic vesicles and the timing of vesicle recruitment to vacancies within the RRP. Second, the alteration was robust across a variety of synapse types embedded within neuronal networks that are subject to dissimilar developmental forces, including the networks

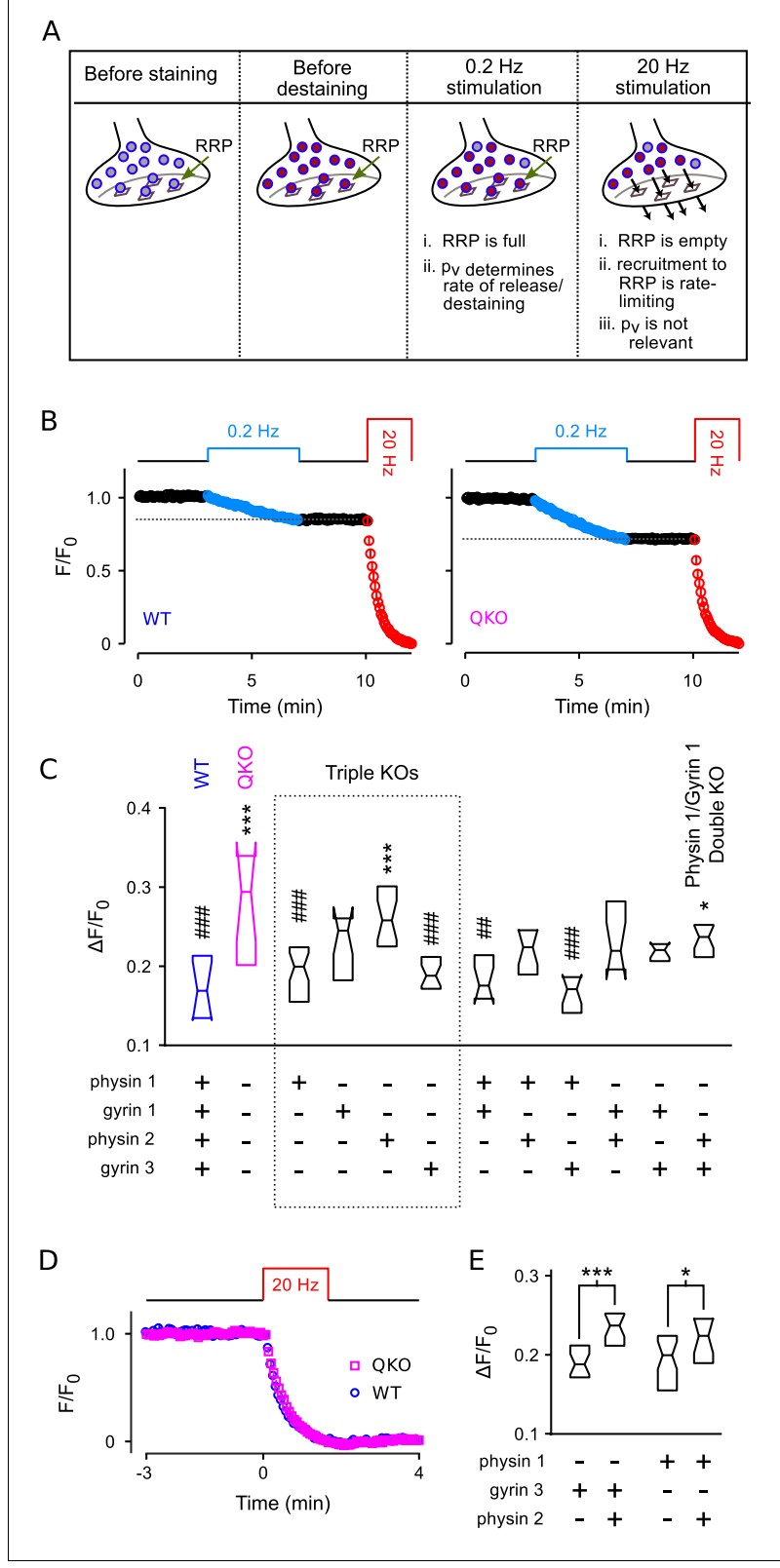

**Figure 7.** Higher throughput analysis of triple and double knockout synapses. (**A**) Diagram of concepts. Recycling vesicles are first stained by driving synaptic vesicle exocytosis and subsequent recycling using electrical stimulation in the presence of FM4-64 in the extracellular fluid. Extracellular dye is then removed and dye stuck to the outside of the plasma membrane is washed off. Fluorescence levels are then monitored with time-lapse imaging as

*Figure 7 continued on next page*

*Figure 7 continued*

synapses are destained by triggering action potentials at low frequency (0.2 Hz). At such a low frequency, the RRP remains almost completely full because the time between action potentials (5 s) is enough for recruitment of new vesicles to replace the ones that undergo exocytosis. Since each action potential releases a higher fraction of the RRP contents at QKO synapses - i.e., because $\bar{p}_v$ is higher - more vesicles undergo exocytosis, and destaining is faster as a consequence. In contrast, recruitment to the RRP becomes rate-limiting during high frequency stimulation that is fast enough to drive the RRP to a near empty steady state (20 Hz). As a consequence, the amount of destaining no longer depends on $\bar{p}_v$, and the synaptophysin family proteins no longer influence the timing. (B) Destaining during electrical stimulation for WT and QKO. Data points are mean ± s.e.m. of median values from each preparation; $n \geq 11$ preparations, each with >250 ROIs. $\Delta F/F_0$ values in (C and E) are calculated as 1 minus the value indicated by the horizontal dashed line. (C) Comparison across genotypes of amount of destaining during the 0.2 Hz train of stimulation. Experimenter was blind to genotype. Boxes are middle two quartiles; horizontal lines are medians; notches signify 95% confidence intervals (\*\*\**p*<0.001, \**p*<0.05, compared to WT; ###*p*<0.001, ##*p*<0.01, compared to QKO; ANOVA followed by Tukey's honest significant difference criterion; $n \geq 11$ for each). (D) No difference between WT and QKO in time course of destaining when the frequency of stimulation was 20 Hz; n ≥ 3 preparations. (E) Follow-on paired tests indicate that synaptophysin 2 lessens the amount of compensation produced by synaptogyrin 3 or synaptophysin 1 when expressed alone (\*\*\**p*<0.001, \**p*<0.05; rank sum).

DOI: https://doi.org/10.7554/eLife.40744.013

The following figure supplement is available for figure 7:

**Figure supplement 1.** Example experiment.

DOI: https://doi.org/10.7554/eLife.40744.014

formed by neurons grown in dissociated cell culture. Third, besides the decrease in VAMP 2, we did not detect substantial changes in levels of a variety of other proteins that have been implicated in exocytosis, including syntaxin 1, SNAP-25, RIM 1/2, Munc13-1, Rab 3a, and complexin 1/2. And fourth, synaptophysin 2 seemed to inhibit the function of other family members, which argues against the specific concern that removing the other family members increased the fusogenicity of synaptic vesicles mechanically, by simply exposing space on the surface vesicular membranes.

Notably, however, no elevation in $\bar{p}_v$ was detected in neuromuscular junctions after deleting the family from *C. elegans* or *Drosophila* - and exocytosis of neurotransmitter was increased rather than decreased after expressing exogenous synaptophysin 1 in *Xenopus* - and it seems unlikely that an entire family of proteins would have an unrelated function in the different species (*Alder et al., 1995*; *Abraham et al., 2006*; *Stevens et al., 2012*). One possibility is that family members function as bi-directional regulators of exocytosis where the directionality is modulated by second messengers, possibly via phosphorylation of the tyrosine residues along the C-terminal tail. In any case, we anticipate that the cause of the discrepancies between species will become clear when more is known about the mechanism.

Indeed, even basic information about the mechanism remains to be elucidated. One possibility is that synaptophysin family members might interact directly with catalysis as outlined in *Rothman et al. (2017)*; see also *Adams et al. (2015)*. However, when bound to synaptophysin 1, VAMP 2 was excluded from the core SNARE complex consisting of VAMP 2, syntaxin 1, and SNAP25 that catalyzes exocytosis (*Edelmann et al., 1995*), and the combination of syntaxin 1 and SNAP25 could disrupt the binding between synaptophysin 1 and VAMP 2 in an enriched vesicle preparation (*Siddiqui et al., 2007*). A second possibility is that family members might lessen SNARE complex formation simply by restricting the availability of VAMP 2. However, we are not aware of evidence that exocytosis can be modulated in this manner, and indeed, lowering levels of the SNAP-25 component paradoxically increased $\bar{p}_v$ in at least one study (*Antonucci et al., 2013*).

In any case, the relevance of the reduction in VAMP 2 levels in purified QKO synaptosomes to the increase in exocytosis seen at intact synapses in the functional assays is not known, and is counter-intuitive given that VAMP 2 is necessary for exocytosis. Intriguingly, the reduction is in-line with a previous study where exogenous VAMP 2 could be driven to synaptic vesicles by co-expressing synaptophysin 1 (*Pennuto et al., 2003*). The results do not suggest that synaptophysin family members are required for targeting VAMP 2 to synaptic vesicles, but are consistent with the possibility that VAMP 2 exists in two pools within vesicle membranes: one of which is stabilized by binding to synaptophysin family members; and the other by a different factor that remains to be identified. On the

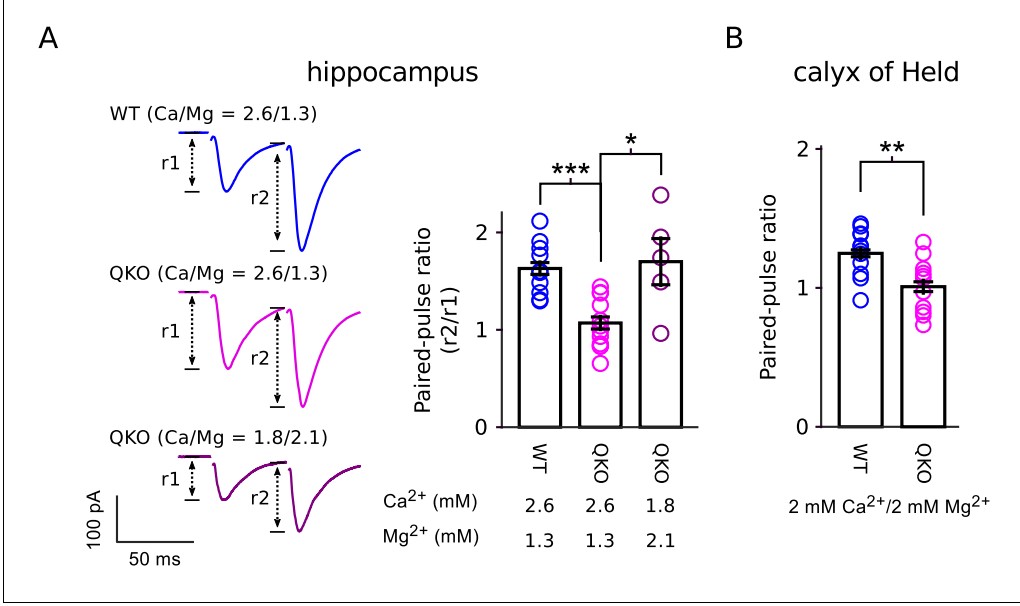

**Figure 8.** Reduced paired-pulse facilitation at QKO synapses is caused by occlusion. (**A**) Schaffer collateral synapses of the hippocampus. Traces are the average of the first two responses during 20 Hz stimulation across the entire data set (inter-pulse interval was 50 ms; n = 8 preparations for WT in 2.6 mM $Ca^{2+}$/1.3 mM $Mg^{2+}$; n = 12 for QKO in 2.6 mM $Ca^{2+}$/1.3 mM $Mg^{2+}$; and n = 5 for QKO in 1.8 mM $Ca^{2+}$/2.1 mM $Mg^{2+}$; * signifies $p < 0.05$, *** is $p < 0.001$; Wilcoxon rank sum; bars are mean ± s.e.m.). 1.8 mM $Ca^{2+}$/2.1 mM $Mg^{2+}$ was chosen for these exeriments because the paired-pulse ratio at QKO synapses then matched WT synapses when bathed in 2.6 mM $Ca^{2+}$/1.3 mM $Mg^{2+}$. QKO synapses exhibited even more paired-pulse facilitation when $Ca^{2+}$ was lowered and/or $Mg^{2+}$ increased further. (**B**). Calyx of Held synapses also exhibited significantly less paired-pulse facilitation in the experiments documented in *Figure 3* (** signifies $p < 0.01$; Wilcoxon rank sum; bars are mean ± s.e.m.). .
DOI: https://doi.org/10.7554/eLife.40744.015

other hand, it is possible that the reduction in QKO synaptosomes resulted from depletion from plasma membrane rather than synaptic vesicles because significant amounts of VAMP 2 are consistently found in plasma membranes (*Sankaranarayanan et al., 2000*), and synaptosomes contain plasma membrane in addition to synaptic vesicles.

It is not clear how or if the small elevation in size of spontaneous responses seen both here and after deleting synaptogyrin from *Drosophila* is related to the elevated probability of release (*Stevens et al., 2012*). We cannot rule out a postsynaptic mechanism, but vesicles were larger in the *Drosophila* mutants, which might play a role. Possibly also relevant: Vesicles were also larger in synapses from knockouts of other presynaptic proteins involved in exocytosis, including VAMP 2, SNAP-25, and Munc13-1/2; the increase in radius was <10%, but even this small increase would translate to an increase in volume of 30%, which is more than the elevation in quantal size seen here (*Imig et al., 2014*).

Our results are in-line with previous reports of a role for synaptophysin family members in short-term synaptic plasticity (*Janz et al., 1999*; *Kwon and Chapman, 2011*; *Rajappa et al., 2016*). However, the decreased paired-pulse facilitation and increased paired-pulse depression seen in single, double and quadruple knockouts at a variety of synapse types do not necessarily indicate defects in mechanisms underlying short-term plasticity, but instead may result from the baseline elevation of $\bar{p}_v$. And indeed, paired-pulse facilitation could be unmasked at QKO synapses by lowering extracellular $Ca^{2+}$, which lowers the baseline (*Figure 8*). In any case, the elevation in $\bar{p}_v$ at QKO synapses cannot be attributed solely to the *superpriming* phenomenon proposed in *Taschenberger et al. (2016)* because the frequency jump experiments documented in *Figure 5* indicated that the elevated release probability pertained to both low and high $p_v$ vesicles, whereas superpriming is thought to pertain exclusively to vesicles with high $p_v$.

The evidence against substantial deficits in the timing of vesicle recruitment to the RRP at QKO synapses seems to be strong. In particular, receptor desensitization mechanisms that could occlude differences between QKO and WT in the electrophysiological studies were ruled out with additional experiments in *Figure 3—figure supplement 1* for calyx of Held and in *Wesseling et al. (2002)* for Schaffer collateral synapses. And, the evidence in *Figure 7D* relies on logic that avoids postsynaptic mechanisms altogether. However, our results do not conflict with the evidence for biochemical and cell biological alterations previously seen downstream of exocytosis in synaptophysin 1 knockouts (*Kwon and Chapman, 2011*; *Gordon et al., 2011*; *Rajappa et al., 2016*), although the results do indicate that any such downstream alterations would not affect the timing of vesicle recruitment to the RRP.

The selectivity of the increase in $\bar{p}_v$, with no change in the timing of vesicle recruitment or the size of the RRP, suggests that synaptophysin family members normally dampen synaptic connection strength when synapses are used at low frequencies. However, their effective role during the type of burst firing that occurs routinely in vivo would be more complex owing to slower depletion of the RRP as a direct consequence of reduced transmitter release. One way to characterize this sort of functional complexity is so called *redistribution of synaptic efficacy* where a decrease in synaptic strength at the beginning of a train of action potentials serves to enhance the strength later on (*Markram and Tsodyks, 1996*). Intriguingly, it seems that some activity-dependent form of redistribution of synaptic efficacy can be induced at a broad range of synapse types. For example, long-lasting bidirectional redistribution can be induced at cortical and hippocampal synapses by some of the same experimental protocols used to activate standard long-term potentiation and depression mechanisms at other synapse types (*Markram and Tsodyks, 1996*; *Sjöström et al., 2003*; *Yasui et al., 2005*; *Monday et al., 2018*). And, although the terminology was different, other reports have suggested that the mechanisms underlying post-tetanic potentiation and a third type of short-term plasticity termed augmentation have a similar re-distributive effect (*Stevens and Wesseling, 1999*; *Garcia-Perez and Wesseling, 2008*; *Lee et al., 2010*). Going forward it will be interesting to determine if synaptophysin family members or other selective regulators of $\bar{p}_v$ such as GIT 1/2 or Mover are involved in any of these phenomena (*Körber et al., 2015*; *Montesinos et al., 2015*).

## Materials and methods

**Key resources table**

| Reagent type (species) or resource | Designation | Source or reference | Identifiers | Additional information |
|---|---|---|---|---|
| Chemical compound, drug | DL-APV | Abcam | Cat# ab120271 | 50 or 100 µM |
| Chemical compound, drug | DNQX | Abcam | Cat# ab120169 | 10 µM |
| Chemical compound, drug | kynurenic acid | Sigma | Cat# K3375 | 1-4 mM |
| Chemical compound, drug | picrotoxin | Sigma | Cat# P1675 | 50 µM |
| Chemical compound, drug | strychnine | Abcam | Cat# ab120416 | 0.5 µM |
| Chemical compound, drug | Advasep-7 | Cydex | Cat# ADV7-03A-03105 | 1 mM |
| Chemical compound, drug | FM4-64 | Biotium | Cat# BT70021 | 15 µM |
| Strain, strain background (mouse) | RRID: IMSR_JAX:008454 | Jackson | Cat# 008454 | CRE expressor |
| Strain, strain background (mouse) | RRID: IMSR_JAX:008415 | Jackson | Cat# 008415 | Syp KO/Syngr1 KO/ Synpr KI/Syngr3 KI |
| Antibodies | | | All | see *Table 1* |

Knockout and WT control mice were obtained from two independent crosses. For the experiments in *Figures 1–5* and *7*, QKO, matched WT controls, and the ten logically possible double and triple knockouts were bred out in three or four generations by crossing a germline CRE expressing line (Jackson labs catalog number 008454) with Jackson line 008415, which carries targeted knockouts of synaptophysin 1 and synaptogyrin 1 genes and floxed conditional mutations of synaptophysin 2 and synaptogyrin 3; the CRE transgene was eliminated during the process. QKO and WT mice for experiments in *Figure 6* were obtained directly from Dr. Thomas Südhof.

## Western blotting

Synaptosomes were prepared as in *Kohansal-Nodehi et al. (2016)*. Six tissue preparations were analyzed in parallel from separate cohorts of five 3-month-old males and females, and a mixed cohort of six 17-day-old males and females. Samples from all six preparations were run on each blot, and each was replicated at least three times. Optical densities were first normalized by the mean density of all six samples, and then re-normalized so that the mean WT value was 1.0 before calculating the summary statistics. Hippocampal tissue homogenates were prepared as in *Fiuza et al. (2013)*. Eight tissue preparations were analyzed in parallel from four males of each genotype. Samples from all eight were run on each blot, and each was replicated three times. Optical densities were normalized by the mean density of the WT samples on each blot, and then values for each sample were averaged across blots before calculating the summary statistics. See *Table 1* for information about antibodies.

## Electrophysiology

Methods were the same as *Mahfooz et al. (2016)* for the calyx of Held and *Wesseling et al. (2002)* for Schaffer collateral synapses. All experiments were done in ex vivo slices from 13- to 21-day-old animals. Unless otherwise noted, n-values in Results and Figure Legends refer to number of preparations; multiple trials from individual preparations were averaged at the level of raw data before further analysis. Experiments in *Figures 1–3*, *5* and *7* were done blind to genotype.

## Cell culture assay

Mice of the various genotypes became available sporadically over a period of seven months. Minimum desired sample sizes of 10 preparations were estimated beforehand based on results from pilot experiments conducted on WT and QKO synapses using 1 Hz rather than 0.2 Hz stimulation. However, actual sample sizes were larger in most cases owing to repetitions conducted to evaluate reproducibility over time.

### FM4-64 fluorescence imaging

Hippocampal neurons were cultured from mice up to one day after birth and grown on glass coverslips coated with laminin and polyornithine as described in *Chowdhury et al. (2013)*. Imaging was performed between 14–21 days after plating on an inverted microscope *via* a 25× oil immersion objective (Zeiss LD LCI Plan-APOCHROMAT 440842–9870; NA = 0.8) using a CCD camera (Photometrics CoolSNAP HQ; on chip binning by 2; pixel size was 0.5 μm × 0.5 μm). Illumination was < $160\,lm$ for 25 ms with a green LED (520 nm; Luxeon LXHL-LM5C) *via* the XF102-2 filter set from Omega Optical. Time lapse imaging was at 0.25 Hz. The imaging chamber was low volume (~35μl) and sealed on top and bottom. Flow was continuous during imaging (0.2-0.5 ml/min). Electrical stimulation was bipolar (0.5 ms at - 30 V then 0.5 ms at + 30 V; Falco Systems WMA 280) *via* two platinum wires (1 mm diameter, separated by ~0.5 cm) that were glued within the chamber and flattened by milling so that the entire lower surface would make contact with the surface of the culture bearing coverslip. A thin layer of already hardened Sylgard 184 (Dow Corning; < 1 mm) was used instead of rubber or vacuum grease for sealing the chamber. FM4-64 was used at 15 μm and loaded with 60 s of 20 Hz stimulation followed by 2 min rest, and then at least 5 min wash in the absence of FM4-64 and presence of 1 mM Advasep-7. Advasep-7 was continuously present during the destaining phase of experiments. Other solutes were (in mM): NaCl (118); KCl (2); $Ca^{2+}$ (2.6); $Mg^{2+}$ (1.3); Glucose (30); and HEPES (25). Neurotransmitter receptors were blocked with (in μM): picrotoxin (50); DNQX (10); and DL-APV (50).

**Table 1.** Antibodies

| Protein | Designation | Source | Species | Dilution | synapto-somes | hipp. tissue |
|---|---|---|---|---|---|---|
| Synaptophysin 1 | RRID:AB_2313839 | Millipore | mouse | 1:2000 | | ✓ |
| | Cl7.2 | *Jahn et al., 1985* | mouse | 1:1000 | ✓ | |
| Synaptophysin 2 | RRID:AB_887841 | Synaptic Sys. | rabbit | 1:1000 | ✓ | |
| | | | | 1:2000 | | ✓ |
| Synaptogyrin 1 | RRID:AB_887818 | Synaptic Sys. | rabbit | 1:2000 | | ✓ |
| | | | | 1:1000 | ✓ | |
| Synaptogyrin 3 | RRID:AB_2619752 | Synaptic Sys. | rabbit | 1:1000 | ✓ | ✓ |
| Synaptotagmin 1 | RRID:AB_10622660 | Enzo | mouse | 1:2000 | | ✓ |
| | RRID:AB_11042457 | Synaptic Sys. | rabbit | 1:1000 | ✓ | |
| Synapsin 1 | RRID:AB_2619772 | Synaptic Sys. | mouse | 1:5000 | | ✓ |
| VAMP 2 | RRID:AB_887811 | Synaptic Sys. | mouse | 1:1000 | | ✓ |
| | | | | 1:2000 | ✓ | |
| RIM1/2 | RRID:AB_887774 | Synaptic Sys. | rabbit | 1:1000 | ✓ | |
| Munc13-1 | RRID:AB_887733 | Synaptic Sys. | rabbit | 1:1000 | ✓ | |
| Complexin1/2 | RRID:AB_887709 | Synaptic Sys. | rabbit | 1:1000 | ✓ | |
| vATPase | RRID:AB_887696 | Synaptic Sys. | rabbit | 1:500 | ✓ | |
| Syntaxin 1 | RRID:AB_887844 | Synaptic Sys. | mouse | 1:1000 | ✓ | |
| $\beta$-actin | RRID:AB_11042458 | Synaptic Sys. | rabbit | 1:1000 | ✓ | |
| | RRID:AB_476744 | Sigma | mouse | 1:10000 | | ✓ |
| SNAP-25 | RRID:AB_2315340 | Synaptic Sys. | mouse | 1:1000 | ✓ | |
| Rab3a,b,c | Cl42.1 | *Matteoli et al., 1991* | mouse | 1:1000 | ✓ | |
| vGAT | SA5387 | *Takamori et al., 2000* | rabbit | 1:500 | ✓ | |
| vGlut 1 | RRID:AB_2187690 | Santa Cruz | goat | 1:1000 | | ✓ |
| | Shigeo3 | *Takamori et al., 2001* | rabbit | 1:2000 | ✓ | |
| vGlut 2 | Shigeo6 | *Takamori et al., 2001* | rabbit | 1:1000 | ✓ | |

DOI: https://doi.org/10.7554/eLife.40744.016

## Processing

Images from time lapse experiments were aligned using the imagej plugin StackReg:Translation (*Thévenaz et al., 1998*) and in house software. Regions of interest (ROIs) were 2 × 2 pixels (1 μm X 1 μm) and were detected with in house software based on the change in contrast during the experiment (see *Figure 7—figure supplement 1*).

## Normalization

For comparing images across preparations, median or individual ROI values were: (1) divided by the mean value of the background region; and then, (2) corrected for any rundown by subtracting the straight line fitting the values during the rest period immediately preceding the 20 Hz train stimulation. Next: (3) $F_{\infty}$ - the residual fluorescence remaining after the final 20 Hz train - was subtracted; and (4) the values were normalized by dividing by $F_0$, which was the mean value over the 2 Min preceding electrical stimulation.

# Acknowledgements

We thank Daniela Urribarri for genotyping the animals and other technical assistance, Dr. Thomas Südhof for providing the equipment, animals, and reagents for the experiments conducted in hippocampal slices, Dr. Santiago Canals for help with the statistical analysis, and Drs. Joan Galcerán, Juan Lerma, Donald Lo, Rafa Fernández-Chacón, and Dani Gitler for suggestions about how to write the manuscript and Dr. David Litvin for suggestions about the writing and help with illustrations.

# Additional information

## Competing interests

Reinhard Jahn: Reviewing editor, *eLife*. The other authors declare that no competing interests exist.

## Funding

| Funder | Grant reference number | Author |
|---|---|---|
| Ministerio de Ciencia y Tecnología | BFU2009-12160 | John F Wesseling |
| Ministerio de Ciencia y Tecnología | SEV-2013-0317 | Isabel Perez-Otano John F Wesseling |
| Universidad de Navarra | | Isabel Perez-Otano John F Wesseling |
| Ministerio de Educación, Cultura y Deporte | Salvador de Madariaga Visiting Scholarship | John F Wesseling |
| Jeronimo de Ayanz program | | John F Wesseling |
| Ministerio de Ciencia y Tecnología | BFU2016-80918R | John F Wesseling |
| Ministerio de Ciencia y Tecnología | SAF2013-48983R | Isabel Perez-Otano John F Wesseling |

The funders had no role in study design, data collection and interpretation, or the decision to submit the work for publication.

## Author contributions

Mathan K Raja, Sergio Del Olmo-Cabrera, Investigation, Writing—review and editing; Julia Preobraschenski, Formal analysis, Investigation, Methodology, Writing—review and editing; Rebeca Martinez-Turrillas, Investigation, Methodology, Writing—review and editing; Reinhard Jahn, Formal analysis, Supervision, Investigation, Methodology, Writing—review and editing; Isabel Perez-Otano, Formal analysis, Investigation, Writing—original draft, Writing—review and editing; John F Wesseling, Conceptualization, Resources, Data curation, Software, Formal analysis, Supervision, Funding

acquisition, Validation, Investigation, Methodology, Writing—original draft, Project administration, Writing—review and editing

### Author ORCIDs
Mathan K Raja https://orcid.org/0000-0001-7727-6994
Reinhard Jahn https://orcid.org/0000-0003-1542-3498
John F Wesseling http://orcid.org/0000-0002-7565-2594

### Ethics
Animal experimentation: Animal protocols were approved by the Universidad de Navarra and Universidad Miguel Hernandez Institutional Animal Care and Use Committees (2017/VSC/PEA/00196) and conformed to the guidelines of Spanish Royal Decree 1201/2005.

### Decision letter and Author response
Decision letter https://doi.org/10.7554/eLife.40744.021
Author response https://doi.org/10.7554/eLife.40744.022

## Additional files

### Supplementary files
• Transparent reporting form
DOI: https://doi.org/10.7554/eLife.40744.017

### Data availability
All data analyzed for this study are included in the manuscript and supporting files. Source data files are available at Dryad (doi:10.5061/dryad.rn91r08).

The following dataset was generated:

| Author(s) | Year | Dataset title | Dataset URL | Database and Identifier |
|---|---|---|---|---|
| Raja MK, Martinez-Turrillas R, del Olmo-Cabrera S | 2018 | Data from: Elevated synaptic vesicle release probability in synaptophysin/gyrin family quadruple knockouts | https://dx.doi.org/10.5061/dryad.rn91r08 | Dryad Digital Repository, 10.5061/dryad.rn91r08 |

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
