## [Decision Letter]

[**Editorial note:** This article has been through an editorial process in which the authors decide how to respond to the issues raised during peer review. The Reviewing Editor's assessment is that all the issues have been addressed.]

Thank you for submitting your article "Synaptophysin/gyrin family proteins are selective negative regulators of exocytosis at mouse central synapses" for consideration by *eLife*. Your article has been reviewed by three peer reviewers, including Graeme W Davis as the Reviewing Editor and Reviewer #1, and the evaluation has been overseen by Gary Westbrook as the Senior Editor. The following individual involved in review of your submission has also agreed to reveal his identity: Nils Brose (Reviewer #2).

The Reviewing Editor has highlighted the concerns that require revision and/or responses, and we have included the separate reviews below for your consideration. If you have any questions, please do not hesitate to contact us.

Summary:

Raja et al. present an interesting story on tetraspan vesicle membrane proteins of the synaptophysin and synaptogyrin families. Synaptic vesicles (SVs) contain three types of tetraspan membrane proteins, synaptophysins, synaptogyrins, and SCAMPs. Although synaptophysins are among the most abundant SV proteins, their function – and the function of synaptophysins, synaptogyrins, and SCAMPs in general – has remained enigmatic. Indeed, genetic elimination of synaptophysin, synaptogyrin, and SCAMP in *C. elegans*, which expresses only one isoform of each family of tetraspan vesicle membrane proteins, has no effect on spontaneous or evoked synaptic transmission (Abraham et al., 2006). Raja et al. performed a related study on mammalian synaptophysins and synaptogyrins, focusing on quadruple KO mice lacking all synaptophysins and synaptogyrins. The key finding is that the loss of synaptophysins and synaptogyrins causes increased SV release probability and faster excitation-secretion coupling. This is clearly interesting and important for the synapse biology field. Raja et al. based their analysis on electrophysiological recordings at the calyx of Held and Schaffer collateral synapses in the hippocampus, and on FM4-64 loading/destaining experiments with cultured hippocampal neurons. Overall, the experiments were done in a stringent manner and the data look convincing. After extensive discussion, all three reviewers agree that the study is an important contribution, but there remain issues that should be addressed prior to publications, as cited below.

Major revisions:

1) The reviewers discussed the issue of interpreting synaptic vesicle fusion in the absence of four synaptic vesicle proteins that are among the most abundant synaptic vesicle proteins. The data shown in Figure S1 show trends regarding changes in SV proteins in the whole hippocampus of quadruple KOs. The issue of compensatory changes in synaptic vesicle proteins as a primary or secondary mediator of the observed effects on synaptic vesicle release is important, and is something that has been routinely addressed in similar knockout studies. As such, the authors could extend this analysis of synaptic vesicle protein abundance in purified synaptic vesicle fractions from the quadruple knockouts, including proteins directly involved in release probability such as RIM, Munc13, Rab3, and Complexin. Alternatively, the authors must provide additional discussion, citing the relevant literature, regarding how any observed changes may contribute, or not contribute, to the observed phenotype. In particular, discuss whether removal of a large fraction of synaptic vesicle protein might directly contribute to enhanced release through direct effects on vesicle fusogenicity.

2) The reviewers recommend either addressing synaptic vesicle endocytosis with direct assays, or altering the text of the manuscript to be more circumspect.

3) It is important that the authors cite and discuss the relevance of work in the *C. elegans* system (Abraham et al., 2006), and discuss more precisely their claim of having assessed a broad range of synapses. Several synapses have been assessed. How similar or distinct are these synapses?

4) It was recommended that the Discussion section be focused more clearly on the data presented. The authors are, of course, free to speculate. However, there was discussion regarding whether the proposed link between the present study and phosphorylation of synaptophysins and synaptogyrins, mover, GITs, ELKS, or tomosyn was a bit large a leap.

5) Authors estimate quanta/vesicle numbers from the charge of EPSCs. In high frequency trains, receptor desensitization, saturation and spill was reported. How does it influence the results?

6) The description of methods and analysis is very limited to the extent that it is e.g. not understandable how the cumulative plots are generated. Why are there steps of different sizes on the y-axes and multiple data points at the same y value?

7) The authors should discuss the following issues:

7A) It remains confusing what vesicles the authors refer to with "eager" and "reluctant". "Eager" is neither referenced nor explained. They claim that low Pr and high Pr vesicles are immediately releasable. Does this mean in parallel? How do the authors exclude that high frequency increases the rate of conversion from low to high Pr? Is it a sequential process before fusion?

7B) How does the reduced delay between AP and vesicle release affect the trains at 300Hz? Is e.g. desensitization, receptor saturation differently affected?

7C) The traces in Figure 4C from the end of the 50Hz train and the switch to 300Hz for the two genotypes look similar to me. It remains unclear how the authors derive a difference in the low Pr vesicles.

7D) Why would high Pr vesicles influence the time course of an EPSC?

7E) A similar amplitude after the 4th AP during a train is consistent with the description of superprimed vesicles (e.g. Taschenberger et al., 2016). Thus in the QKO, more vesicles could reside in the superprimed state. An estimate of the RRP from the steady state at the end of a train does not allow to distinguish between vesicle numbers in different Pr states.

Minor Comments:

1) In subsection "Higher throughput assay in primary cell culture" the authors refer the reader to a figure legend for the explanation of the authors' interpretation of the de-staining data. This explanation should be given in the main text.

2) The description of methods and analysis is very limited to the extent that it is e.g. not understandable how the cumulative plots are generated. Why are there steps of different sizes on the y-axes and multiple data points at the same y value?

Separate reviews (please respond to each point):

*Reviewer #1:*

The authors generate quod knockout mice, deleting Synaptophysin 1 and 2 and synaptogyrin 1 and 3. Subsequent analyses at Calyx and hippocampal synapses in the quad-KO demonstrate altered presynaptic vesicle release consistent with an increase in presynaptic release probability. The fact that this effect is observed at two very different synapses, in vitro and in vivo, argues for the generality of the observed effects. The paper ends with a modest amount of data on gene-specific contributions, highlighting a potentially interfering function for synaptophysin 2. In general, the experiments seem to have been performed to a high standard, are clearly presented and appropriately discussed. The findings are clear, and provide what could be considered a definitive set of experiments regarding the required action of this set of proteins for synaptic transmission. Given that very similar phenotypes are observed at Calyx and hippocampus, in vivo and in vitro, it seems unlikely that the effects will be caused by altered developmental trajectory of synapse development. However, this was never directly addressed. There is a curious effect on synaptic delay at the Calyx. It would be nice to know if this was altered at the hippocampal synapses. My only other comment is that an assessment of intrinsic excitability would be easily achieved in both systems, somatic in the hippocampus and presynaptically at the Calyx. Given that there is a shortened delay in the calyx and there is evidence of increased excitability, I am curious if the action potential waveform is altered. While this would be expected to alter the waveform of the EPSC, it remains a final piece of the puzzle that could be resolved. Given that these studies are carried out in quad KO animals, I am hesitant to require additional work unless it provides fundamental evidence in favor of the existing data, or has the potential to alter the fundamental conclusions in some manner.

*Reviewer #2:*

Raja et al. present an interesting story on tetraspan vesicle membrane proteins of the synaptophysin and synaptogyrin families. Synaptic vesicles (SVs) contain three types of tetraspan membrane proteins, synaptophysins, synaptogyrins, and SCAMPs. Although synaptophysins are among the most abundant SV proteins, their function – and the function of synaptophysins, synaptogyrins, and SCAMPs in general – has remained enigmatic. Indeed, genetic elimination of synaptophysin, synaptogyrin, and SCAMP in *C. elegans*, which expresses only one isoform of each family of tetraspan vesicle membrane proteins, has no effect on spontaneous or evoked synaptic transmission (Abraham et al., 2006). Raja et al. performed a related study on mammalian synaptophysins and synaptogyrins, focusing on quadruple KO mice lacking all synaptophysins and synaptogyrins. The key finding is that the loss of synaptophysins and synaptogyrins causes increased SV release probability and faster excitation-secretion coupling. This is clearly interesting and important for the synapse biology field.

Raja et al. based their analysis on electrophysiological recordings at the calyx of Held and Schaffer collateral synapses in the hippocampus, and on FM4-64 loading/destaining experiments with cultured hippocampal neurons. Overall, the experiments were done in a stringent manner and the data look convincing. Nevertheless, I have several comments regarding the data and their interpretation that the authors should address.

1) In principle, I like the mouse KO approach. However, when four proteins are removed from SVs – and among these some of the most abundant SV proteins – the issue of 'collateral damage' arises. In essence, I think the authors can at present not be sure that the quadruple KO SVs are properly equipped with all relevant protein components. The data shown in Figure S1 already show trends of changes in SV proteins in the whole hippocampus of quadruple KOs. To rule out substantial changes at the level of SV composition, it would be necessary to systematically assess protein levels in purified SV fractions from quadruple KOs. The problem here may be that not many labs can do this properly.… but it could be done within a reasonable time frame on the basis of a collaboration (e.g. with people like Reinhard Jahn). If this is out of the question, the authors should at least discuss this issue and tone down the manuscript text regarding collateral changes in SV composition after loss of all synaptophysins and synaptogyrins. I personally think that SV composition may well be changed, contributing to the phenotypical changes observed.

2) An issue related to the one above (point 1) concerns the intrinsic 'fusogenicity' of SVs after loss of all synaptophysins and synaptogyrins. Upon removal of these proteins, one has to assume that large SV surface areas are stripped of a 'protein coat', so that much more 'naked' lipid membrane is exposed than in wild-type SVs. What argues against the possibility that this alone changes the fusion characteristics of SVs in a manner that might explain the phenotypes observed (e.g. because no 'protein coat' is present to inhibit the interactions of the SV and target membrane, thereby facilitating fusion)? I think this issue should at least be discussed.

3) In subsection "No alteration in the timing of vesicle trafficking", the authors relate their findings to earlier claims that synaptophysin 1 may play a role in post-endocytic clearing of SV material from the presynaptic membrane. In this context, they cite three papers, two of which mainly addressed a role of Synaptophysin in endocytosis without focusing on release-site clearance, which are related but different processes. I think the data presented in the present study cannot be taken as strong evidence against a role of synaptophysins and synaptogyrins in endocytosis. To make this claim, the authors would have to assess endocytosis directly. If this is out of the question, the corresponding text should be toned down.

4) In the Abstract and the main text, the authors state that they studied "a broad range of synapse types". I find this a bit hyperbolic. They studied calyx of Held synapses, Schaffer collateral synapses, and synapses of cultured hippocampal neurons. This is nice – I do not want to be misunderstood here – but not a "broad range". After all, some calyx experts have repeatedly made the casual statement that the calyx of Held is like a parallel array of hippocampal-like synapses, and Schaffer collateral synapses in slices and synapses in cultured hippocampal neurons are not so dissimilar either.

5) Based on the argument above, I am not convinced that the authors' "comparison between calyx of Held and Schaffer collateral synapses is a good test for the generality [of the phenotypes they observe] across synapse types". On the contrary, I think the general relevance of the present findings remains questionable. This is particularly critical in view of a corresponding study on *C. elegans* (Abraham et al., 2006), which showed that the complete loss of synaptophysin, synaptogyrin, and SCAMP has no effect on spontaneous or evoked synaptic transmission and which is not even cited in the present study. I think that not citing the study by Abraham et al., 2006, is a problematic omission that must be rectified. Further, I think that in the course of discussing the Abraham-paper the whole 'generality' discussion should be toned down in view of the arguments I made above.

6) The discussion part of the present paper is too speculative for my taste. The link between the present study and phosphorylation of synaptophysins and synaptogyrins, mover, GITs, ELKS, or tomosyn feels a bit constructed and arbitrary. After all, several other mutants show increased vesicular release probability. Likewise, the discussion of "activity-dependent redistribution of synaptic efficacy" goes too far for my taste; at present, there is no connection. Instead, I suggest to discuss the issues mentioned above, which directly relate to the actual findings of the present study.

Minor Comments:

In subsection "Higher throughput assay in primary cell culture" the authors refer the reader to a figure legend for the explanation of the authors' interpretation of the de-staining data. I think this explanation should be given in the main text.

*Reviewer #3:*

This is an interesting study by Raja and colleagues to revisit the function of the synaptic vesicle protein families of synaptogyrins and synaptophysins. Using a combination of multiple KO mice, they achieved a step forward in uncovering one of their physiological functions.

The function of these protein families remained largely illusive and controversial. Overexpression previously showed an inhibitory role of synaptogyrins and synaptophysin on release in PC12 cells. Here, authors confirm this function and report that these proteins act at different synapses in the CNS in a similar way. Authors report that isoforms function partially redundantly and a full functional loss is only achieved with certain deletion of multiple isoforms. They describe an interesting effect on the delay between action potentials and vesicle release as well as provide a robust analysis of changes in release probability, while vesicle pools sizes appear unchanged. However, the further dissection of the mechanism remains inconclusive.

– that bind via VAMP 2 to the machinery that catalyzes exocytosis." Misleading statement. They might bind to VAMP 2 before it engages into the fusion machinery.

– It remains confusing what vesicles the authors refer to with "eager" and "reluctant". "Eager" is neither referenced nor explained. They claim that low Pr and high Pr vesicles are immediately releasable. Does this mean in parallel? How do the authors exclude that high frequency increases the rate of conversion from low to high Pr? Is it a sequential process before fusion?

– Why would high Pr vesicles influence the time course of an EPSC?

– A similar amplitude after the 4th AP during a train is consistent with the description of superprimed vesicles (e.g. Taschenberger et al., 2016). Thus in the QKO, more vesicles could reside in the superprimed state. An estimate of the RRP from the steady state at the end of a train does not allow to distinguish between vesicle numbers in different Pr states.

– The traces in Figure 4C from the end of the 50Hz train and the switch to 300Hz for the two genotypes look similar to me. Not clear to me how the authors derive a difference in the low Pr vesicles.

– Some proteins were analyzed, but not proteins previously shown to regulate release probability, such as RIM, Munc13, Rab3, Complexin. Are those proteins upregulated and mediate the increase in Pr?

– The description of methods and analysis is very limited to the extent that it is e.g. not understandable how the cumulative plots are generated. Why are there steps of different sizes on the y-axes and multiple data points at the same y value?

– Authors estimate quanta/vesicle numbers from the charge of EPSCs. In high frequency trains, receptor desensitization, saturation and spill was reported. How does it influence the results?

– How does the reduced delay between AP and vesicle release affect the trains at 300Hz? Is e.g. desensitization, receptor saturation differently affected?

---

## [Author Response]

Reviewer #1:

*The authors generate quod knockout mice, deleting Synaptophysin 1 and 2 and synaptogyrin 1 and 3. Subsequent analyses at Calyx and hippocampal synapses in the quad-KO demonstrate altered presynaptic vesicle release consistent with an increase in presynaptic release probability. The fact that this effect is observed at two very different synapses,* in vitro *and* in vivo*, argues for the generality of the observed effects. The paper ends with a modest amount of data on gene-specific contributions, highlighting a potentially interfering function for synaptophysin 2. In general, the experiments seem to have been performed to a high standard, are clearly presented and appropriately discussed. The findings are clear, and provide what could be considered a definitive set of experiments regarding the required action of this set of proteins for synaptic transmission. Given that very similar phenotypes are observed at Calyx and hippocampus,* in vivo *and* in vitro*, it seems unlikely that the effects will be caused by altered developmental trajectory of synapse development. However, this was never directly addressed. There is a curious effect on synaptic delay at the Calyx. It would be nice to know if this was altered at the hippocampal synapses. My only other comment is that an assessment of intrinsic excitability would be easily achieved in both systems, somatic in the hippocampus and presynaptically at the Calyx. Given that there is a shortened delay in the calyx and there is evidence of increased excitability, I am curious if the action potential waveform is altered. While this would be expected to alter the waveform of the EPSC, it remains a final piece of the puzzle that could be resolved. Given that these studies are carried out in quad KO animals, I am hesitant to require additional work unless it provides fundamental evidence in favor of the existing data, or has the potential to alter the fundamental conclusions in some manner.*

We agree with the reviewer on all of these points and appreciate the input very much.

The fact that we analyzed a variety of synapse types embedded within neuronal networks that are subject to dissimilar developmental forces (including networks formed by neurons grown in dissociated cell culture) is now mentioned in the second paragraph of the Discussion.

We thought the comments about the synaptic delay at the calyx of Held (Figure 1D of the initial submission) were especially insightful and we agree that this merits a better analysis. Instead, though, we have now removed the panel. The original experiments were done blind to genotype. However, the effect was not reproduced in a second data set designed specifically to address the present comments, possibly owing to an unexpected amount of variation between preparations. At this point, we would like to be cautious because we believe the decreased time – if truly occurring downstream of Ca^2+^ influx – may have important implications for understanding the coupling between Ca^2+^ and how exocytosis is catalyzed. We do not have the resources to do this well in a short amount of time, but plan to follow this up later. In any case, we have measured action potential conduction times along the axon in the calyx of Held preparation, and did not find any difference between WT and QKO.

Reviewer #2:

Raja et al. present an interesting story on tetraspan vesicle membrane proteins of the synaptophysin and synaptogyrin families. Synaptic vesicles (SVs) contain three types of tetraspan membrane proteins, synaptophysins, synaptogyrins, and SCAMPs. Although synaptophysins are among the most abundant SV proteins, their function – and the function of synaptophysins, synaptogyrins, and SCAMPs in general – has remained enigmatic. Indeed, genetic elimination of synaptophysin, synaptogyrin, and SCAMP in *C. elegans*, which expresses only one isoform of each family of tetraspan vesicle membrane proteins, has no effect on spontaneous or evoked synaptic transmission (Abraham et al., 2006). Raja et al. performed a related study on mammalian synaptophysins and synaptogyrins, focusing on quadruple KO mice lacking all synaptophysins and synaptogyrins. The key finding is that the loss of synaptophysins and synaptogyrins causes increased SV release probability and faster excitation-secretion coupling. This is clearly interesting and important for the synapse biology field.Raja et al. based their analysis on electrophysiological recordings at the calyx of Held and Schaffer collateral synapses in the hippocampus, and on FM4-64 loading/destaining experiments with cultured hippocampal neurons. Overall, the experiments were done in a stringent manner and the data look convincing. Nevertheless, I have several comments regarding the data and their interpretation that the authors should address.

We thank the reviewer for insightful comments, and pointing out key areas where we were naive. We have now followed essentially all of the recommendations except a single one about “redistribution of synaptic efficacy” as outlined below.

1) In principle, I like the mouse KO approach. However, when four proteins are removed from SVs – and among these some of the most abundant SV proteins – the issue of 'collateral damage' arises. In essence, I think the authors can at present not be sure that the quadruple KO SVs are properly equipped with all relevant protein components. The data shown in Figure S1 already show trends of changes in SV proteins in the whole hippocampus of quadruple KOs. To rule out substantial changes at the level of SV composition, it would be necessary to systematically assess protein levels in purified SV fractions from quadruple KOs. The problem here may be that not many labs can do this properly.… but it could be done within a reasonable time frame on the basis of a collaboration (e.g. with people like Reinhard Jahn). If this is out of the question, the authors should at least discuss this issue and tone down the manuscript text regarding collateral changes in SV composition after loss of all synaptophysins and synaptogyrins. I personally think that SV composition may well be changed, contributing to the phenotypical changes observed.

We have now collaborated with Dr Jahn to analyze a larger variety of synaptic proteins. We used purified synaptosomes instead of purified synaptic vesicles to save time and to simultaneously analyze presynaptic proteins that are not integrated into vesicle membranes. However, purified synaptosomes are thought to provide an accurate readout for the relative abundance of synaptic vesicle proteins (Wilhelm et al., 2014).

We did see ~50% reduction in VAMP 2 levels, in-line with a trend in our original analysis, and consistent with a significant reduction seen previously for synaptophysin 1 knockouts (McMahon et al., 1996). This is intriguing and may be related to the phenotype. If so, however, the mechanism is not obvious because VAMP 2 is required for exocytosis. This is now discussed in the Discussion section (fifth paragraph).

Besides the decrease in VAMP 2, we did not see alterations in: other synaptic vesicle proteins including synaptotagmin 1, vGlut 1&2, and vATPase; or in other proteins previously implicated in regulating release as recommended by the third reviewer (i.e., SNAP25, syntaxin1, RIM1/2, Munc13-1, Complexin, and Rab3).

2) An issue related to the one above (point 1) concerns the intrinsic 'fusogenicity' of SVs after loss of all synaptophysins and synaptogyrins. Upon removal of these proteins, one has to assume that large SV surface areas are stripped of a 'protein coat', so that much more 'naked' lipid membrane is exposed than in wild-type SVs. What argues against the possibility that this alone changes the fusion characteristics of SVs in a manner that might explain the phenotypes observed (e.g. because no 'protein coat' is present to inhibit the interactions of the SV and target membrane, thereby facilitating fusion)? I think this issue should at least be discussed.

The results in Figure 7E suggesting that synaptophysin 2 competitively inhibits the function of synaptophysin 1 and synaptogyrin 3 would seem to argue against the possibility. This is now mentioned explicitly in the Discussion (end of second paragraph).

3) In subsection "No alteration in the timing of vesicle trafficking", the authors relate their findings to earlier claims that synaptophysin 1 may play a role in post-endocytic clearing of SV material from the presynaptic membrane. In this context, they cite three papers, two of which mainly addressed a role of Synaptophysin in endocytosis without focusing on release-site clearance, which are related but different processes. I think the data presented in the present study cannot be taken as strong evidence against a role of synaptophysins and synaptogyrins in endocytosis. To make this claim, the authors would have to assess endocytosis directly. If this is out of the question, the corresponding text should be toned down.

We agree and have now adjusted the text accordingly. We continue to acknowledge that our results do not contradict the results of endocytosis experiments on synaptophysin 1 knockouts, but we removed the speculation about an alternative interpretation (eighth paragraph of Discussion).

4) In the Abstract and the main text, the authors state that they studied "a broad range of synapse types". I find this a bit hyperbolic. They studied calyx of Held synapses, Schaffer collateral synapses, and synapses of cultured hippocampal neurons. This is nice – I do not want to be misunderstood here – but not a "broad range". After all, some calyx experts have repeatedly made the casual statement that the calyx of Held is like a parallel array of hippocampal-like synapses, and Schaffer collateral synapses in slices and synapses in cultured hippocampal neurons are not so dissimilar either.

We have now replaced “broad range” with “variety”.

We agree that calyx of Held synapses are similar at the level of first principles to others with active zones. In fact, we published evidence supporting this in Mahfooz et al., 2016. However, our point then was not that there are no differences between synapses. Indeed, we saw order of magnitude sized differences in vesicle recruitment rates, and baseline mean p_v_ was approximately triple at calyx of Held compared to Schaffer collaterals. Instead, our point was that the different types of synapses shared the same first principles and that the differences in function would be caused by differences in parameter values rather than by qualitative differences in underlying mechanisms. We believe that comparing Schaffer collateral synapses to calyces of Held is a particularly good test for generality across synapse types in the present context because of the large differences in the timing of vesicle recruitment, which is particularly relevant because a current hypothesis predicts that the timing would be altered in the knockout, and that the defect would be more prominent during heavier use.

5) Based on the argument above, I am not convinced that the authors' "comparison between calyx of Held and Schaffer collateral synapses is a good test for the generality [of the phenotypes they observe] across synapse types". On the contrary, I think the general relevance of the present findings remains questionable. This is particularly critical in view of a corresponding study on *C. elegans* (Abraham et al., 2006), which showed that the complete loss of synaptophysin, synaptogyrin, and SCAMP has no effect on spontaneous or evoked synaptic transmission and which is not even cited in the present study. I think that not citing the study by Abraham et al., 2006, is a problematic omission that must be rectified. Further, I think that in the course of discussing the Abraham-paper the whole 'generality' discussion should be toned down in view of the arguments I made above.

We agree that we were naive on this point. We now cite Abraham et al., 2006, and another study where synaptogyrin was knocked out of *Drosophila*, and discuss both in the third paragraph of the Discussion. Claims about generality are toned down throughout the manuscript.

6) The discussion part of the present paper is too speculative for my taste. The link between the present study and phosphorylation of synaptophysins and synaptogyrins, mover, GITs, ELKS, or tomosyn feels a bit constructed and arbitrary. After all, several other mutants show increased vesicular release probability. Likewise, the discussion of "activity-dependent redistribution of synaptic efficacy" goes too far for my taste; at present, there is no connection. Instead, I suggest to discuss the issues mentioned above, which directly relate to the actual findings of the present study.

We now limit mention of GITs and Mover to the last sentence of the Discussion, with no mention at all of ELKS and tomosyn.

We have preserved mention of activity-dependent re-distribution of synaptic efficacy. We would like to keep this because we think the term is an elegant way to describe the phenotype, and it would seem like an oversight to avoid making the connection.

Minor Comments:In subsection "Higher throughput assay in primary cell culture" the authors refer the reader to a figure legend for the explanation of the authors' interpretation of the de-staining data. I think this explanation should be given in the main text.

The explanation is now included in the main text as recommended.

Reviewer #3:

This is an interesting study by Raja and colleagues to revisit the function of the synaptic vesicle protein families of synaptogyrins and synaptophysins. Using a combination of multiple KO mice, they achieved a step forward in uncovering one of their physiological functions.The function of these protein families remained largely illusive and controversial. Overexpression previously showed an inhibitory role of synaptogyrins and synaptophysin on release in PC12 cells. Here, authors confirm this function and report that these proteins act at different synapses in the CNS in a similar way. Authors report that isoforms function partially redundantly and a full functional loss is only achieved with certain deletion of multiple isoforms. They describe an interesting effect on the delay between action potentials and vesicle release as well as provide a robust analysis of changes in release probability, while vesicle pools sizes appear unchanged. However, the further dissection of the mechanism remains inconclusive.

We thank the reviewer for raising these points, some of which invoke what we believe are key outstanding issues for understanding presynaptic function in general. We believe that we have well-reasoned responses, which are outlined below.

– that bind via VAMP 2 to the machinery that catalyzes exocytosis." Misleading statement. They might bind to VAMP 2 before it engages into the fusion machinery.

We agree that this was poorly worded. It seems that there is a range of thoughts about this, which is now referenced and discussed in the fourth paragraph of the Discussion.

– It remains confusing what vesicles the authors refer to with "eager" and "reluctant". "Eager" is neither referenced nor explained. They claim that low Pr and high Pr vesicles are immediately releasable. Does this mean in parallel? How do the authors exclude that high frequency increases the rate of conversion from low to high Pr? Is it a sequential process before fusion?

a) The term “eager” has now been replaced by the concept of vesicles belonging to a high p_v_ subdivision of the RRP;

b) We use *p_v_* and mean p_v_ instead of Pr to avoid a separate sort of confusion;

c) We do think that low *p_v_* vesicles are immediately releasable, in parallel with high *p_v_* vesicles;

d) We do not exclude conversions between low to high *p_v_* states, or frequency-dependent acceleration of the mechanism (Legend of Figure 5A);

e) We believe that the evidence against key alternate scenarios where synaptophysin family members would modulate transition from an un-releasable to releasable state is strong.

The explanations are below.

We think that low *p_v_* vesicles are immediately releasable because release of neurotransmitter continues to be tightly synchronized to action potentials during frequency jumps that are initiated after the high *p_v_* vesicles have been eliminated (Figure 5B and Figure 5—figure supplement 1C). Moreover, the low *p_v_* vesicles likely are released in parallel with high *p_v_* vesicles during low frequency stimulation when the RRP is full because synapses with more low *p_v_* vesicles express more paired-pulse facilitation whereas synapses with fewer express more paired-pulse depression (Figure 9 of Mahfooz et al., 2016).

We are aware that conversion from low to high *p_v_* states plays a key role in models from other groups, and we do not rule this out – so long as vesicles in the low p_v_ subdivision can be released directly to account for the synchronicity of release immediately after the frequency jump (see Results under heading “Elevated probability of release.…”, and Legend of Figure 5A). Nor do we rule out the idea that high frequency stimulation might accelerate the rate of transfer. In fact, we were able to elucidate a surprising quantitative constraint in Mahfooz et al., 2016; specifically, the transfer between subdivisions would have to be ∼9/s during 300Hz stimulation, at room temperature – not substantially faster or slower – and would have to be reversible. This constraint is not altered at QKO synapses

Notably, our own WT and QKO data could be modeled equally well by eliminating the transfer in both directions. This eliminates a degree of freedom and has interesting consequences for how synapses might be modulated at the level of mechanism (last paragraph of Results of Mahfooz et al., 2016). In addition, at least one of the recent articles proposing a serial scheme acknowledged that the authors could not rule out alternatives where low and high *p_v_* vesicles are processed in parallel (Miki et al., 2016). And finally, there seems to be a growing body of evidence that stable high and low release probability release sites do co-exist at some synapse types (Hu et al., 2013; Muller et al., 2015; Böhme et al., 2016).

Nevertheless, we have considered alternate scenarios where the low *p_v_* vesicles are not actually immediately releasable and the ongoing synchronous transmitter release in the absence of high *p_v_* vesicles is explained by fast transfer of a tiny fraction of the un-releasable vesicles to an immediately releasable state in the time between action potentials. The fraction would be the value of mean p_v_ we estimated for the low *p_v_* vesicles. Our understanding is that the reviewer is asking about the subset of models in this category where the essentially equivalent value for mean p_v_ at 50 and 100Hz in Figure 5E is explained by activity dependent acceleration of the rate of transfer that is close to directly proportional to the frequency of stimulation.

We found that models of this class are not compatible with the results of the frequency jump experiments. We arrived at this conclusion as follows. We reasoned that the acceleration mechanism could not influence the rate of transfer from the un-releasable to immediately releasable state until *after* the first 3.33 ms interstimulus interval during 300Hz stimulation. In contrast, the rate of transfer *during* the first 3.33 ms interstimulus interval would have to be equivalent to the rate during the first 3.33ms of the 20 or 10ms interstimulus intervals during the preceding 50 or 100Hz stimulation. Therefore, one would expect 6-fold or 3-fold fewer new immediately releasable vesicles at the end of the first 3.33ms interstimulus interval compared to before the preceding action potential. If so, the second pulse during 300Hz stimulation would elicit paired-pulse depression rather than the facilitation seen in Figure 5D and Figure supplement 1B. And, indeed, one would expect twice as much paired-pulse depression in Figure 5D compared to Figure supplement 1B (i.e., because the transfer rate during 50Hz stimulation would be half the rate when stimulation is 100Hz). If the acceleration was very fast, the third pulse could then elicit facilitation.This reasoning is now included in the Legend of Figure 5—figure supplement 1.

For completeness: We are aware that it would be possible to rescue the alternate scenario by replacing the premise that activity accelerates the transfer from un-releasable to immediately releasable states with the premise that single action potentials – at any frequency – always induce transfer of a similar fraction of un-releasable vesicles to the immediately releasable state. If so, synaptophysin family members would have two functional roles:

a) Negative regulation of the releasablility of immediately releasable vesicles; and,

b) Negative regulation of the fraction of un-releasable vesicles that are transferred to the immediately releasable state after each action potential.

However, we do not include a discussion of models of this final class in the manuscript because our understanding is that none have yet been proposed, and it seems unlikely to us because the two roles would have analogous functional effects on the releasability of vesicles in the high and low p_v_ subdivisions of the RRP via dissimilar types of mechanisms.

– Why would high Pr vesicles influence the time course of an EPSC?

We are not certain what is meant here. We would not necessarily expect an influence, but such a factor could explain the results in Fedchyshyn and Wang, 2007.

– A similar amplitude after the 4th AP during a train is consistent with the description of superprimed vesicles (e.g. Taschenberger et al., 2016). Thus in the QKO, more vesicles could reside in the superprimed state.

We agree that the plot in Figure 3C is reminiscent of the effect produced by PdBu in Taschenberger et al., 2016. However, our understanding is that the superpriming idea proposed in Taschenberger et al., 2016, supposes that vesicles are converted from the high *p_v_* state to the superprimed state, whereas mean p_v_ at QKO synapses seems to be elevated for both the low and high p_v_ subdivisions at QKO synapses; this is now noted in the seventh paragraph of the Discussion, which now includes a reference to Taschenberger et al., 2016. Note, however, that the plot in Figure 3C is equally reminiscent of Figure 2B of Mahfooz et al., 2016, where mean p_v_ was increased by raising extracellular Ca^2+^, which presumably would not alter the number of superprimed vesicles. Also relevant: Ongoing experiments in the lab suggest that post-tetanic potentiation is intact in QKO synapses; the results will be included in a separate article (we haven’t tried PdBu yet).

An estimate of the RRP from the steady state at the end of a train does not allow to distinguish between vesicle numbers in different Pr states.

We agree and did not mean to suggest otherwise. A key point is that we define mean p_v_ to be equal to the number of release events following an action potential divided by the quantal content of the RRP immediately beforehand; mean p_v_ now replaces some instances of *p_v_* in the previous version to make this more clear. Thus, mean p_v_ is the mean probability of release of all vesicles within the RRP. Our reasoning does not rely on any assumptions about the distribution of release probabilities of the individuals within the RRP.

We reasoned that the mean p_v_-value for the vesicles remaining within the RRP during steady state 50 or 100 Hz stimulation can be calculated by dividing the number of quanta released by each action potential during steady state stimulation by the steady state quantal content of the RRP (Figure 5E) just as mean p_v_ for the vesicles within the RRP after long rest intervals can be calculated by dividing the number of quanta released by the first action potential in a train by the quantal content of the RRP when completely replenished (Figure 3E).

We then found that the mean p_v_-value for vesicles remaining in the RRP during steady state 50 or 100 Hz stimulation was lower than the value when the RRP was full for both genotypes, confirming that at least some of the high *p_v_* vesicles were eliminated. Next, the mean p_v_ value during 50 Hz stimulation was essentially equivalent to during 100 Hz stimulation, even though the standing fullness of the RRP was substantially less during 100 Hz stimulation (Figure 5—figure supplement 2), indicating that almost all of the high *p_v_* vesicles were eliminated during both frequencies of stimulation. Finally, the mean p_v_-values for QKO synapses during 50 and 100 Hz stimulation were higher than the corresponding values for WT synapses (Figure 5E). We interpret these results as indicating that synaptophysin family proteins negatively regulate the probability of release of both low and high *p_v_* vesicles, and as arguing against scenarios where only vesicles in the high p_v_ subdivision are affected.

– The traces in Figure 4C from the end of the 50Hz train and the switch to 300Hz for the two genotypes look similar to me. Not clear to me how the authors derive a difference in the low Pr vesicles.

We have clarified this in the Results and Figure 5F where the data are normalized so that both QKO and WT results can be plotted on the same graph. Most of the difference between QKO and WT is not actually visible in Figure 5C – Figure 4C in the previous version is Figure 5C in the current version – because of how the data are binned; each plot on the graph is the sum of 6 EPSCs. However, one can still see that – even when binned – responses did decay faster at QKO synapses, and reached a new steady state sooner.

– Some proteins were analyzed, but not proteins previously shown to regulate release probability, such as RIM, Munc13, Rab3, Complexin. Are those proteins upregulated and mediate the increase in Pr?

We have now looked at levels of RIM1/2, Mun13-1, Raba,b,c, and Complexin in purified synaptosomes in collaboration with Dr. Reinhard Jahn. The results are in Figure 1 and Figure 1—figure supplement 1. We did not find clear evidence for upregulation of any.

– The description of methods and analysis is very limited to the extent that it is e.g. not understandable how the cumulative plots are generated. Why are there steps of different sizes on the y-axes and multiple data points at the same y value?

The cumulative plots of summary statistics have been replace with standard bar graphs; individual values are plotted on top of the bars. The cumulative plots of electrophysiological responses during repetitive stimulation have not been altered, but these are standard in the field and our understanding is that it was the plots of summary statistics which were seen as difficult to interpret.

– Authors estimate quanta/vesicle numbers from the charge of EPSCs. In high frequency trains, receptor desensitization, saturation and spill was reported. How does it influence the results?

Responses:

a) Desensitization: Our conditions were different from the conditions of studies reporting postsynaptic receptor desensitization and saturation in two regards that we believe are relevant: (1) we used tissue from older animals; and (2), experiments where quantal content is estimated from responses during repetitive stimulation were conducted in the presence of 1mM kynurenic acid. We have now added Figure 3—figure supplement 1A-E to show that receptor desensitization does not play a substantial role under these conditions.

b) Saturation: Likewise, receptor saturation was not a concern in the presence of kynurenic acid because responses were reduced greatly in size, by 85% ± 2% for QKO and 85% ± 3% for WT. Saturation (or voltage clamp errors) could affect estimates of EPSC size recorded in the absence of kynurenic acid when measured as the peak or current integral. However, the estimates of amount of block by 1mM kynurenic acid were likely not affected substantially because we measured the slope of the rising phase (20 to 60%) instead of the peak or current integral for this purpose.

c) Spill: We have had personal conversations with calyx of Held experts who have raised concerns that part of the response we see during steady state 300Hz stimulation might arise from glutamate that spills over from nearby synaptic clefts. Newly added Figure 3—figure supplement 1F now addresses this possibility.

In any case:

i) Our estimates of quantal content and unitary rates of vesicle recruitment during repetitive use agree with estimates from other studies that used different techniques (Neher, 2010);

ii) We think the estimates are accurate, but even if not, the key conclusions are drawn from either the presence or absence of differences between WT and QKO synapses, rather than from the precise values of quantal content or recruitment rates; and,

iii) The key conclusions that involve quantal estimates are additionally demonstrated in two other ways using different techniques and preparations. In particular, the experiments in cell culture in Figure 7D avoid completely the involvement of postsynaptic receptors

– How does the reduced delay between AP and vesicle release affect the trains at 300Hz? Is e.g. desensitization, receptor saturation differently affected?

We have removed the evidence for a reduced delay between AP and vesicle release as explained in the response to the first reviewer. The results in Figure 3—figure supplement 1AE show that receptor desensitization does not seem to play a role in 1mM or more kynurenic acid

References:

Fedchyshyn MJ & Wang LY (2007). Activity-dependent changes in temporal components of neurotransmission at the juvenile mouse calyx of Held synapse. Journal of Physiology 581, 581–602.

Neher E (2010). What is Rate-Limiting during Sustained Synaptic Activity: Vesicle Supply or the Availability of Release Sites. Frontiers in Synaptic Neuroscience 2, 144.